# The rapid and highly parallel identification of antibodies with defined biological activities by SLISY

Steve Lu[1,2], Austin K. Mattox[1,2], P. Aitana Azurmendi[3], Ilias Christodoulou[4], Katharine M. Wright [3], Maria Popoli[1,2], Zan Chen[3], Surojit Sur[1,2], Yana Li[3], Challice L. Bonifant [4], Chetan Bettegowda [1,5], Nickolas Papadopoulos [1,4], Shibin Zhou [1,4], Sandra B. Gabelli [3,4], Bert Vogelstein [1,2] & Kenneth W. Kinzler[1,4] ✉

The therapeutic applications of antibodies are manifold and the emergence of SARS-CoV-2 provides a cogent example of the value of rapidly identifying biologically active antibodies. We describe an approach called SLISY (Sequencing-Linked ImmunoSorbent assaY) that in a single experiment can assess the binding specificity of millions of clones, be applied to any screen that links DNA sequence to a potential binding moiety, and requires only a single round of biopanning. We demonstrate this approach using an scFv library applied to cellular and protein targets to identify specific or broadly reacting antibodies. For a cellular target, we use paired HLA knockout cell lines to identify a panel of antibodies specific to HLA-A3. For a protein target, SLISY identifies 1279 clones that bound to the Receptor Binding Domain of the SARS-CoV-2 spike protein, with >40% of tested clones also neutralizing its interaction with ACE2 in in vitro assays. Using a multi-comparison SLISY against the Beta, Gamma, and Delta variants, we recovered clones that exhibited broad-spectrum neutralizing potential in vitro. By evaluating millions of scFvs simultaneously against multiple targets, SLISY allows the rapid identification of candidate scFvs with defined binding profiles facilitating the identification of antibodies with the desired biological activity.

Antibodies have been used for the treatment of human disease in one form or another for more than a century[1,2]. Over this time, the tools available for these applications evolved from crude sera to molecularly defined and engineered monoclonal antibodies. Today therapeutic antibodies are used to treat a variety of morbidities including enve-noming, infectious disease, cancer and autoimmunity[3].

Recent events have highlighted the potential value of passive antibody administration for prevention and treatment of infectious diseases. In the past two decades, three zoonotic beta-coronaviruses have crossed species and infected humans with high mortality and impact on society. Most recently during the coronavirus disease 2019 (COVID-19) pandemic, the novel severe acute respiratory syndrome-related coronavirus 2 (SARS-CoV-2) has infected over 575 million people with numbers still rising as of August 2022[4]. Despite the remarkable efforts to develop therapies for infected individuals, the persistence of the pandemic as well as the potential for future outbreaks highlight

[1]Ludwig Center, Sidney Kimmel Comprehensive Cancer Center, Johns Hopkins University School of Medicine, Baltimore, MD 21287, USA. [2]Howard Hughes Medical Institute, Chevy Chase, MD 20815, USA. [3]Department of Biophysics and Biophysical Chemistry, Johns Hopkins University School of Medicine, Baltimore, MD 21287, USA. [4]Department of Oncology, Johns Hopkins University School of Medicine, Baltimore, MD 21287, USA. [5]Department of Neurosurgery, Johns Hopkins University School of Medicine, Baltimore, MD 21205, USA. ✉e-mail: kinzlke@jhmi.edu

limitations of current approaches to identify antibody-based therapies against a highly transmissible and rapidly evolving virus.

Antibody-based therapies have been shown to provide immediate passive protection and to complement prophylactic vaccines. A notable example of this principle is convalescent plasma, though limited by scalability[5]. Another example is ansuvimab (mAb114), which provided a safe and effective treatment against symptomatic Ebola virus disease[6,7]. Other examples are provided by monoclonal antibodies against HIV, Dengue, and Middle Eastern Respiratory Syndrome (MERS). Such antibodies have been identified in screens using Epstein-Barr virus-immortalized memory B cells[8–11]. These highly effective approaches are, however, limited by the time required to obtain samples from recovered patients as well as the time required for identification and isolation of specific, high affinity antibodies[12]. Notably, it has been a challenge clinically to keep up with which mAb available for SARS-CoV-2 are still useful against the prevalent circulating variant[13].

In vitro selection using display technologies, such as biopanning with libraries made in phage, bacteria, yeast, mammalian cells, or with ribosomes, enable particularly efficient ways to select for highly specific binding elements[14,15]. Furthermore, promising candidates can be amplified and enriched through repeated rounds of positive selection with the target[16–18]. Another advantage over conventional antibody generation is that display technologies allow for negative biopanning, which reduce clones that may be cross-reactive with closely related target antigens.

Traditional display library screens employing Sanger sequencing limit the number of clones to be screened to $10^2$–$10^3$. Advances in next-generation sequencing (NGS) have allowed for deeper insights into the diversity by providing up to $10^7$ sequences (10,000-fold more)[19–21]. Although the strength of all of these methods relies on a direct link between phenotype and genotype, high throughput methods to assess clones suffer from imperfections in this link. Specifically, selection for high antigen-affinity clones through multiple rounds of biopanning can be confounded by unpredictable, growth-based library biases as well as stochastic sampling of clones that are at low frequency in the library. Moreover, up until now, sequencing of the entire length of some clone inserts (i.e. 600 bp for single-chain variable fragments (scFvs)) has not been possible without the isolation, cloning and sequencing of individual candidates, which drastically limits throughput. Of note, the potential of advances in sequencing to accelerate this process was suggested by the development of alanine scanning, a method developed over two decades ago that combined capillary sequencing of roughly 600 sequences with phage-displayed libraries of proteins substituted with alanine at 19 positions to quickly explore the binding interactions[22]. Within three weeks' time, it determined the functional contributions of multiple side chains buried at the interface of human growth hormone and its receptor. However its utility is limited to studying interactions between known proteins as it requires designing individual libraries tailored for each target. As of this time, methods to rapidly identify highly specific antibodies against multiple known or potentially uncharacterized (e.g. cell surface) targets in parallel are lacking.

Here, we describe an approach that combines NGS with differential binding assays to simultaneously evaluate all clones in any library even after a single round of screening and rapidly proceed to antibody production.

## Results and discussion
### Basic elements of SLISY
Commonly applied phage display-based approaches for identification of antigen-specific scFvs rely on multiple rounds of biopanning. On the basis of extensive experience with this process, we postulated that the direct assessment of scFv phage binding without the intermediate growth phases could accelerate and improve the identification of

desirable scFvs[23–26]. To test this hypothesis, we first developed a quantitative NGS based phage binding assay called SLISY (Sequencing-Linked ImmunoSorbent assaY). SLISY is based on using the highly diverse complementarity determining region H3 (CDR-H3) as a unique identifier of each scFv-expressing phage (Fig. 1a). We designed primers that could universally amplify the CDR-H3 region of our scFv library with high efficiency while also incorporating a unique molecular barcode to avoid introduction of PCR bias (rather than phage growth bias) in subsequent analyses[27]. We could then use NGS to detect and directly enumerate phage binding to a target antigen versus non target or control antigens. The resulting molecular counts could then be used to calculate a SLISY Binding Ratio (SBR) and identify binders as depicted in Fig. 1a.

In the second component of our strategy, we developed a NGS sequencing strategy that allowed isolation of the full-length scFv sequence of each phagemid in our library using only the CDR-H3 region as a key to query a custom *en masse* sequence library of the input phage (Supplementary Fig. 1). Given that our scFv library is based on the humanized 4D5 (trasuzumab) framework generated against ERBB2, we designed three primers against various constant regions in the backbone so that three rounds of sequencing would cover all the variable regions. These sequences could then be used to rapidly reconstitute binding phage or be converted to full-length antibodies.

### Identification of highly-specific HLA-A3 antibody fragments
As an initial test of the above strategy, we sought to identify antibodies that bound specifically to Human Leukocyte Anitgen-A3 (HLA-A3) allele versus HLA-A2. We chose cell-bound HLA as an initial target because of the availability of positive controls and the opportunity to highlight the need for negative selection to achieve specificity in the background of many related molecules on the cell surface. Four rounds of biopanning were performed with positive selection to parental CFPAC pancreatic cancer cells expressing wild-type HLA-A3/HLA-A2 and negative selection against CFPAC cells with HLA-A3 knocked out (Fig. 1b). In a separate proof-of-principle condition, we spiked-in phage clone A3-Clone 20, a known HLA-A3 binder isolated in previous studies, to a final amount of 1:E5 in the starting library to track its performance as a positive control across biopannings[28]. After four rounds, an SBR for every scFv was calculated as the observed read count of its CDR-H3 sequence recovered after binding to the parental CFPAC cells divided by the observed read count of the same CDR-H3 sequence recovered after binding to the HLA-A3 KO cells. To measure enrichment after four rounds, we calculated a Panning Enrichment Ratio (PER) for each clone by dividing its fraction in the population after growth by the fraction present in the input library and then converting the value to log base 2.

To validate the potential of SLISY to rapidly identify highly specific clones, we selected a panel of 11 candidate clones considering that ideal clones theoretically would have high SBR values as well as enrich over time, especially relative to nonspecific clones (Fig. 1c, Supplementary Data 1). Consistent with our expectations, the CDR-H3 representative of A3-Clone 20 yielded an SBR significantly greater than 1 (7.02 ± 0.42, mean ± SD, $n = 3$) in the sample derived from the spike-in library. Given that A3-Clone 20 as a positive control had an SBR of 7.02, we were interested in selecting clones with a higher SBR (>10) under the assumption that clones with at least a 10-fold theoretical specificity as detected by SLISY would be sufficient for validation. Interestingly, the A3-Clone 20 was only detected in the spike-in library and only five of the high scoring scFvs overlapped between the two biopannings, suggesting the wide diversity of clones available in the library for HLA binding and supporting our concerns about the stochastic and limited nature of growth-based panning.

As described above, we directly generated scFv-expressing phage of our selected clones by sequencing the entire length of scFvs (Supplementary Data 2). When expressed on phagemids, all 11 scFvs reliably

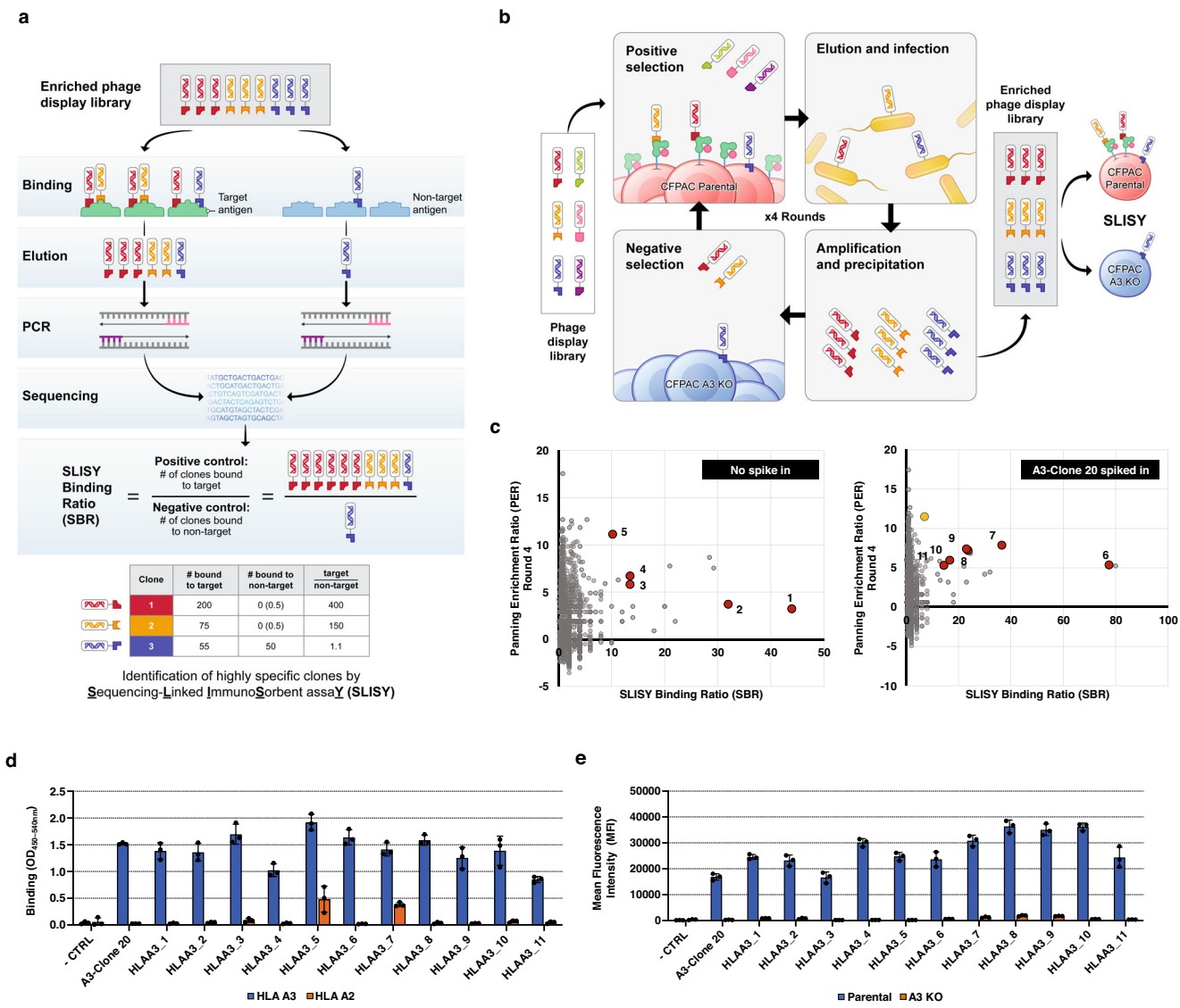

**Fig. 1 | Overview of Sequencing-Linked ImmunoSorbent assaY (SLISY).**
**a** Schematic of SLISY. A sample of scFv phage library is applied to the desired target as well as an appropriate negative control. After extensive washing, bound phage are eluted, followed by PCR amplification and sequencing of the CDR-H3 region with universal primers to determine SLISY Binding Ratios (SBR) for each particular phage clone. SBR is calculated by comparing the number of phage eluted from the target well/cells with identical CDR-H3 sequence to the phage eluted from the non-target (negative) well/cells with the same CDR-H3 sequence. CDR-H3 sequences that have a molecular count of zero are given a value of 0.5. **b** Schematic of biopanning for HLA-A3 binders. The scFv phage display library was expanded and applied to CFPAC parental (HLA-A2 & HLA-A3 positive) cells for positive

biopanning and CFPAC HLA-A3 knockout (KO) cells for negative biopanning against HLA-A2 and other cell surface proteins. **c** SBR versus Panning Enrichment Ratio (PER) for all clones (circles) in HLA-A3 biopanning after four rounds of biopanning. Eleven clones (red) with SBR greater than 10 were selected for subsequent validation. The clone in yellow represents A3-Clone 20. **d** Binding of candidate clones to biotinylated HLA-A3 versus HLA-A2. **e** Binding of clones to CFPAC parental cells that express HLA-A3 versus isogenic HLA-A3 KO cell line. - CTRL represents the background binding of secondary antibodies without any initial phage. Error bars represent standard deviation of means (n = 3). Source data are provided as a Source Data file.

displayed preferential HLA-A3 binding relative to HLA-A2 by enzyme-linked immunosorbent assay (ELISA) and flow cytometry on CFPAC cells (Fig. 1d, e). Interestingly, 7 of 11 clones had a frequency less than 0.0001% in the final pool (Supplementary Data 1). If we picked and characterized a thousand clones, the approximate probability of isolating any given one of these 7 clones ranged from 0.01 to 0.09 and would be only 3E-10 for identifying all 7 in any one sampling. These initial experiments indicated that SLISY could identify desirable clones at very low frequency that would be difficult to identify using a growth based screening approach. The results of the cell-based screening for HLA-A3 suggested that SLISY could improve both the number and quality of clones derived from biopanning and that the SLISY approach warranted further investigation.

**Identification of highly specific SARS-CoV-2 antibody fragments**
Given the importance of SARS-CoV-2 and its emerging variants, we next tested whether SLISY could rapidly identify antibody fragments that specifically bound to the SARS-CoV-2 spike protein. We selected for phage binding to the full-length (FL) SARS-CoV-2 spike protein (amino acids 16 to 1213), to its S1 subunit (S1, amino acids 16-685), or to the core receptor binding domain (RBD, amino acids 319-541) (Fig. 2a). Four rounds of biopanning were performed against each target with negative selection against the related coronavirus MERS (Fig. 2b). After four rounds of biopanning (see below for results of a single round of biopanning), we observed 8312 unique scFv clones with SBR > 10 for the full-length spike protein, S1, or RBD domains (Supplementary Fig. 2a). Of the subset of clones that were selected for binding to the

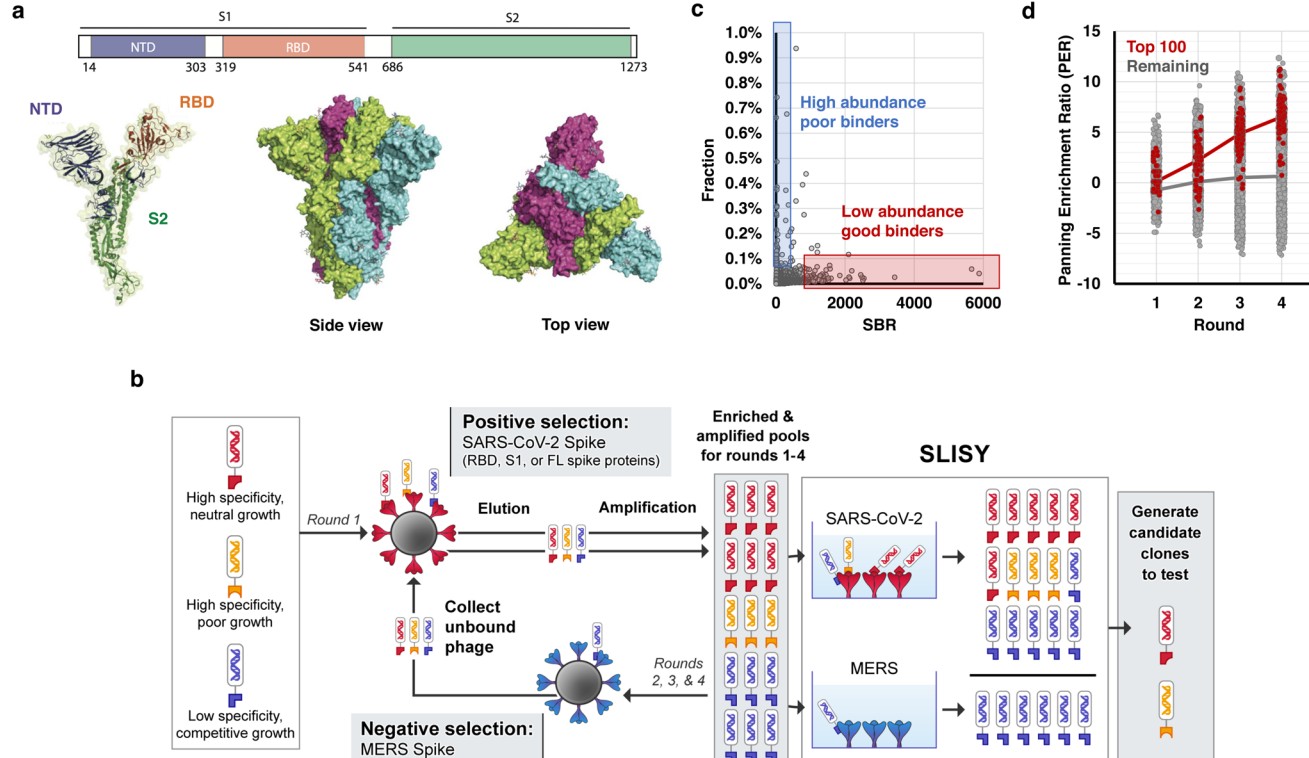

**Fig. 2 | Applying SLISY to SARS-CoV-2 spike protein. a** SARS-CoV-2 trimer. Scheme of the SARS-CoV-2 spike protein with ribbon and surface representations of the SARS-CoV-2 monomer showing the NTD, RBD and S2 domains (PDB ID "6VSB").[26] **b** Schematic of biopanning and SLISY for SARS-CoV-2 specific scFvs. **c** Low representation of clones with high SBR. Fraction of each clone within the entire pool after four rounds of panning against the RBD is plotted against its SBR. Red box indicates clones that are likely good binders (high SBR) but at low abundance. Blue box indicates clones that are poor binders (low SBR) but at high abundance. **d** Enrichment of clones across multiple rounds of biopanning. The top 100 ranked clones were determined based on their SBR values at round 4. The red line represents average enrichment of the round 4 top 100 clones; the grey line represents average enrichment of remaining clones. Source data are provided as a Source Data file.

full-length spike protein, 32% bound to the RBD region of S1 and 7% bound to the non-RBD portion of S1, while the remainder (61%) bound to other regions of the FL protein (Supplementary Fig. 2b).

As observed with HLA-A3 biopanning, many of the most specific SARS-CoV-2 clones (defined by SLISY) were at very low frequency even after four rounds of enrichment and expansion. Conversely, the great majority of the most abundant clones were not highly specific (Fig. 2c). For example, the clone with the highest SBR was only present in one of 2437 clones (0.04%). If clones were selected for evaluation by conventional methods (i.e., a few hundred clones picked randomly after four rounds), this highly specific clone would have been missed. The top 100 clones as assessed by SLISY only represented 1.1 to 4.0% of the final pools for each of the three targets after four rounds (Supplementary Data 3). Although only a small final fraction, clones that scored highest by SLISY were continually enriched across each round (Fig. 2d).

### Validation of the specificity of SARS-CoV-2 clones identified with SLISY

To validate the specificity of the clones identified by SLISY in an orthogonal fashion, we directly generated candidate scFv-expressing phage by sequencing the entire length of scFvs as described above. For validation, we chose 39 clones, thirteen from each of the three proteins used for screening with each clone having an SBR > 100 (Fig. 3a). One-hundred percent of these 39 clones were found to specifically bind to SARS-CoV-2 via ELISA, a standard immunoassay (Fig. 3b). This perfect concordance is not unexpected in that SLISY can be thought of as a type of high-throughput digital ELISA assay in that it evaluates millions of clones in parallel and can detect differences at the level of single phagemids[29].

To be therapeutically useful, an antibody must neutralize the virus rather than simply bind to the SARS-CoV-2 spike protein. Indeed, most antibodies against the SARS-CoV-2 spike protein are not neutralizing because they do not bind to the region required to interact with angiotensin converting enzyme 2 (ACE2), the cellular receptor for the virus[30]. We used an FDA-approved test (cPASS) originally developed to identify neutralizing antibodies to evaluate the capacity of the 39 phage clones described above to block the binding of ACE2 to the spike protein[31]. Over 40% of clones were found to be inhibitory, often as strongly blocking as commercial antibodies (Fig. 3c).

### SLISY allows identification of clones after a single round of biopanning

Although we employed four rounds of biopanning for the experiments described above, we suspected that functional clones could be obtained after one round of biopanning. Moreover, we believed that additional rounds might be unnecessary and decrease the diversity of identified clones by virtue of growth bias: highly specific clones might grow less well than other clones during phage expansion, which is a known problem with biopanning[32]. The most definitive of experiments to address this issue was a comparison of the SLISY data after one versus four rounds of biopanning. We found that 46% of the 100 most specific clones in Round 4 were found among the 100 most specific clones in Round 1 (Fig. 3d). Conversely, 76% of the 100 most specific clones in Round 1 were found within the 250 most specific clones in Round 4 (Fig. 3e). Importantly, these data suggest that neither sequential positive selection nor negative selection is required during the biopanning process when SLISY is employed; biopanning against MERS proteins was not used for Round 1 and only used for Rounds 2, 3,

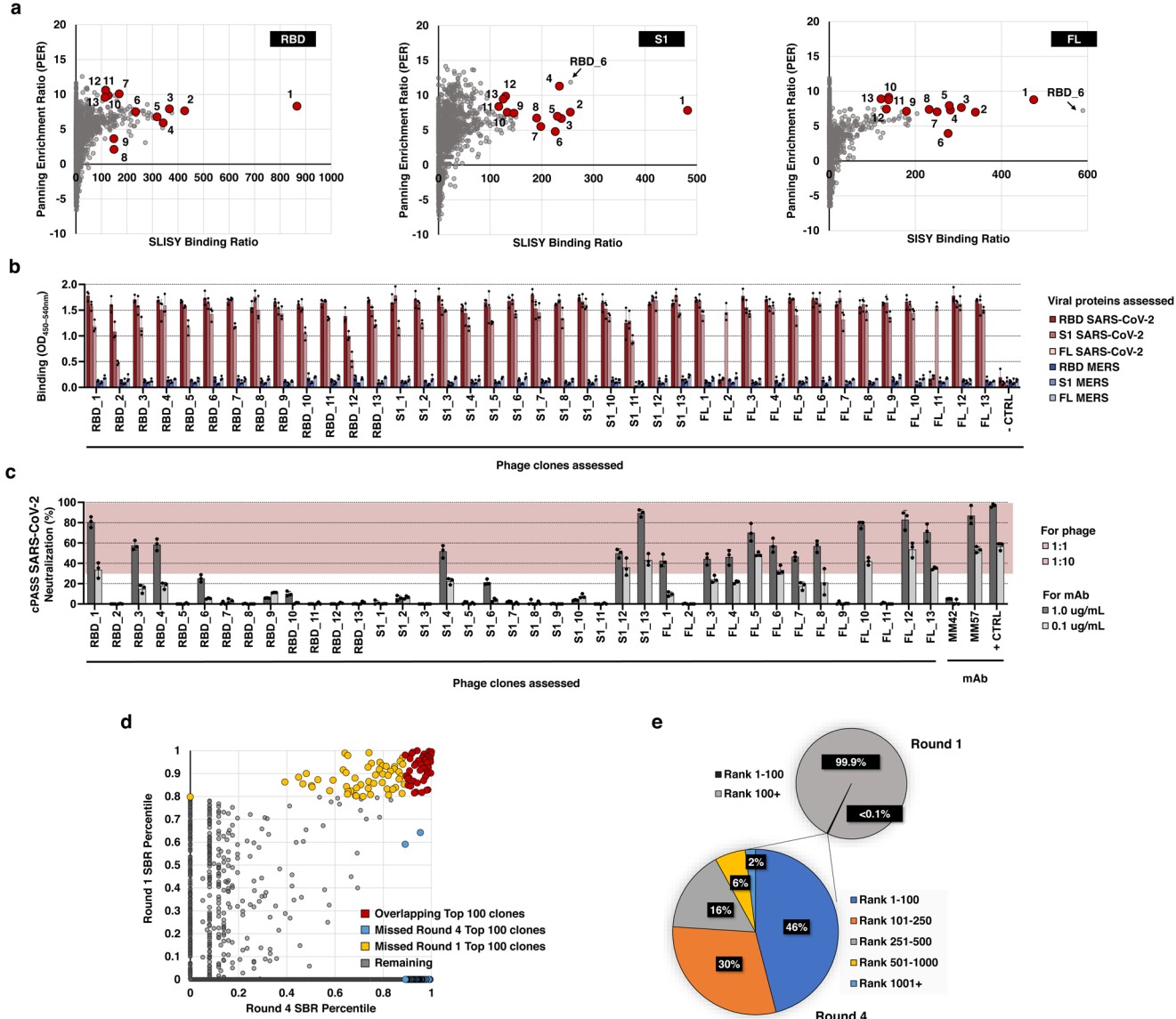

**Fig. 3 | Validation of highly specific SARS-CoV-2 clones. a** SBRs for each clone is plotted against its PER from baseline after four rounds of biopanning. Red clones represent those selected for validation. Note one clone, RBD_6, was highly enriched in all three biopanning strategies but only selected as an RBD candidate as indicated. **b** Binding specificity of candidate SARS-CoV-2 scFvs. Wells pre-coated with RBD, S1 or FL spike proteins from SARS-CoV-2 or MERS were incubated with phage. - CTRL, a negative control using phage expressing scFv against HLA-A3. Error bars represent standard deviation of means (*n* = 3). **c** Neutralizing potential of SARS-CoV-2 specific phage tested using FDA-approved cPASS assay. Phage (10¹³ titer) or antibodies pre-mixed with HRP-RBD were applied to wells pre-coated with recombinant human ACE2 protein. Red region represents ≥30% neutralization

compared to HRP-RBD alone. MM42, a non-neutralizing negative control mAb; MM57, a neutralizing positive control mAb; + CTRL, a manufacturer-provided positive control. Error bars represent standard deviation of means (*n* = 3).
**d** Correlation of SBR between round 4 and round 1 of RBD panning using 100X deeper sequencing. Clones in red represent those that were in the top 100 for both round 1 and round 4. Clones in blue represent those that were in top 100 of round 4 but not round 1. Clones in yellow represent those that were in top 100 of round 1 but not round 4. **e** Performance of clones selected after one round of biopanning. Clones in the top 100 of round 1 after deep sequencing were assessed for their final outcomes after four rounds of biopanning. Source data are provided as a Source Data file.

and 4. This makes the entire biopanning process considerably simpler, quicker and allows identification of clones specific for multiple antigens in a single step, as subsequent experiments confirmed.

### Full-length monoclonal antibodies (mAbs) against SARS-CoV-2 derived from phage clones

To evaluate whether the results described above are due to the scFv itself and independent of the phage structure, we converted the scFv regions from 12 of the phage described above (eight neutralizing and four non-neutralizing) into full-length IgG antibodies, which are more suitable for clinic use (Fig. 4a)[33]. All twelve were successfully made into full-length mAbs (examples in Supplementary Fig. 3a–g) and retained

specific binding to the SARS-CoV-2 spike protein that was equivalent to its phage counterpart (Figs. 3b, 4b). Moreover, the blocking activity was fully retained in six of the eight mAbs, with activities comparable to that of a commercially available, highly neutralizing monoclonal antibody (Fig. 4c, d). It is possible that for the two clones (RBD_3 and S1_4) for which the phage but not the full-length antibodies were able to block ACE2 binding, steric factors associated with the larger phage interfered with binding of the spike protein to ACE2 but were eliminated in the much smaller antibody format. To test if these six mAbs are indeed neutralizing, we incubated HEK-293T cells stably expressing ACE2 simultaneously with mAbs and lentivirus carrying a luciferase coding plasmid and pseudotyped with the SARS-CoV-2 spike

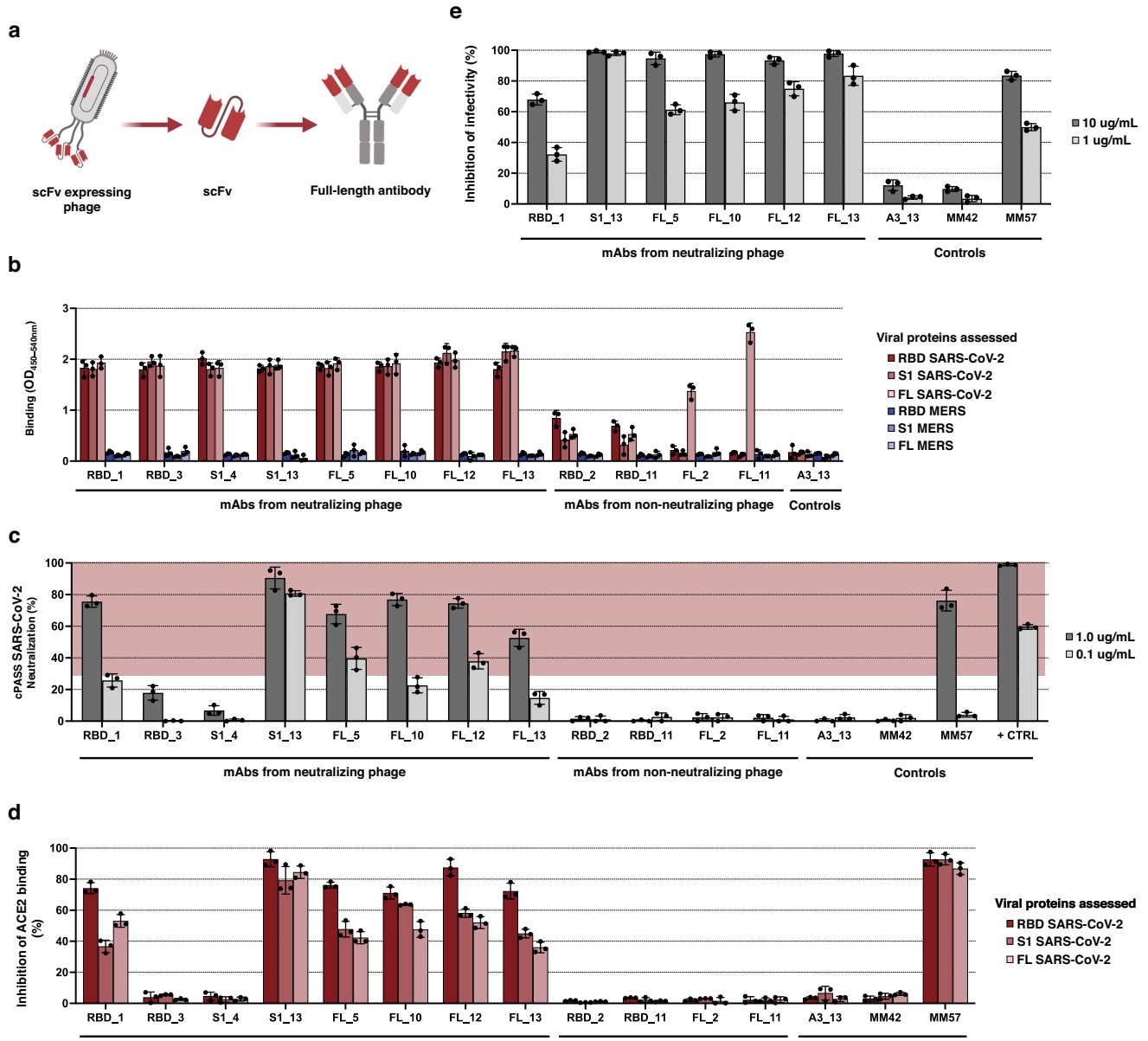

**Fig. 4 | Characterization of converted full-length SARS-CoV-2 antibodies.**
**a** Schematic of relationship between scFv-expressing phage and full-length anti-bodies. Created with BioRender.com. **b** Specificity of full-length antibodies grafted from validated scFv-expressing phage. Wells pre-coated with RBD, S1 or FL spike proteins from SARS-CoV-2 or MERS were incubated with converted antibodies. A3_13, which binds to HLA-A3, serves as a negative control for the ELISA. **c** Measuring neutralization with cPASS assay. Red shaded region represents ≥30% neutralization and is defined as detection of SARS-CoV-2 neutralizing activities. MM42, a non-neutralizing negative control mAb. MM57, a neutralizing positive

control mAb. + CTRL, a manufacturer-provided positive control. **d** Ability of full-length antibodies to inhibit ACE2-spike protein interaction. The converted full-length antibodies were applied to wells pre-coated with RBD, S1 or FL spike proteins from SARS-CoV-2, followed by the addition of recombinant ACE2-His protein. **e** Ability of full-length antibodies to block pseudovirus infectivity. Converted full-length antibodies were incubated with pseudovirus expressing SARS-CoV-2 spike protein and ACE2-expressing HEK-293T cells for 48 h. Infectivity was measured by luciferase activity. Error bars represent standard deviation of means (n = 3). Source data are provided as a Source Data file.

glycoprotein and mAbs for 48 hrs. All six mAbs blocked infectivity as measured by lack of luciferase expression comparable to or better than a commercially available neutralizing mAb indicating that SLISY is able to identify strongly neutralizing mAbs (Fig. 4e).

### Biophysical characterization and epitope mapping of full-length antibodies
Kinetic and equilibrium constants for the six neutralizing mAbs were measured using Surface Plasmon Resonance (SPR). None of them appeared to bind to the control MERS spike protein, as expected, but all of them bound to the SARS-CoV-2 FL spike protein with high affinity

- $K_D$s in the low nanomolar range (6 to 44.2 nM) (Supplementary Fig. 3h, Supplementary Data 4).

Cocktails composed of more than one neutralizing antibody have been shown to have advantages over a single antibody in prior studies[34–36]. To evaluate the potential for such combinations with the new mAbs described here, each was labeled with biotin and competed with each of the others for binding to an immobilized RBD protein (Supplementary Fig. 4). When two mAbs bind to non-overlapping regions of the RBD, there should be little or no decrease in binding of the first mAb by the second mAb. This enabled the grouping of our six neutralizing mAbs into two groups: Group A (RBD_1, RBD_3) and Group

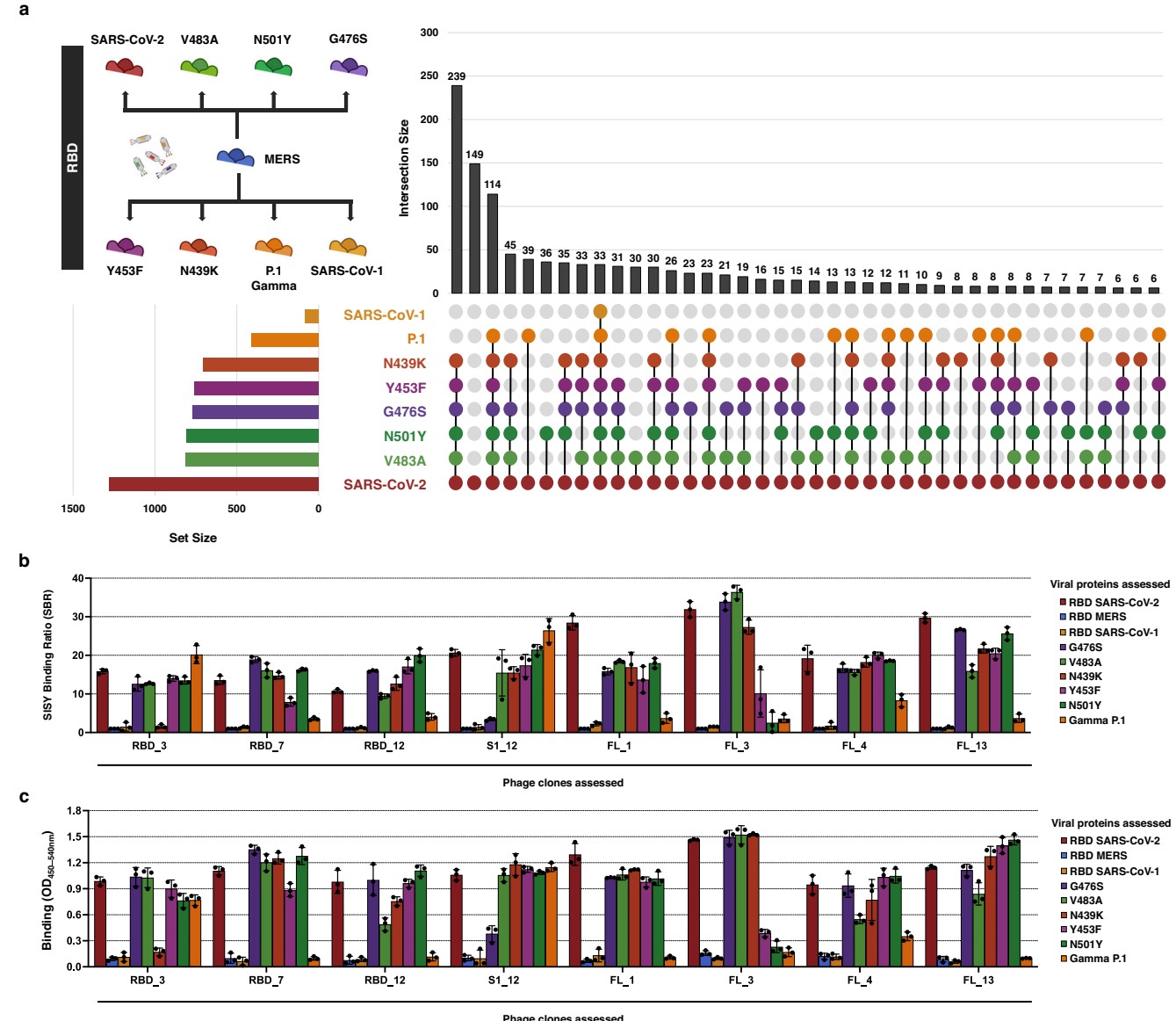

**Fig. 5 | Assessment of binding across multiple variants simultaneously using SLISY. a** Assessing clones across multiple RBD variants. UpSet plot includes upright bars that represent the total number of cross-reactive clones in each intersection. Filled circles below bars represent the variants that represent the intersecting set. Intersecting sets that have fewer than 6 cross-reactive clones were not represented. Horizontal bars represent the total number of clones that had a SLISY Ratio of ten and above for each of the proteins. Created with BioRender.com. **b** SBR for selected clones across variants. **c** ELISA for selected clones across variants. Error bars represent standard deviation of means ($n = 3$). Source data are provided as a Source Data file.

B (FL_5, FL_12, FL_13, S1_13). Binding of a commercially available non-neutralizing, control RBD antibody was not affected by any of the six neutralizing antibodies, and vice versa.

### Assessment of binding to SARS-CoV-2 variants

Rather than perform multiple individual ELISAs for each clone that was sequenced, we asked whether SLISY can be used to rapidly identify and predict patterns of binding across multiple variants of SARS-CoV-2. To address this question, we used the phage that survived four rounds of biopanning against the original SARS-CoV-2 RBD region and performed SLISY against five variants, each containing a single amino acid change in the RBD, and a pathogenic variant with multiple mutations (Gamma P.1) that emerged during the pandemic. Without any further rounds of biopanning, we found that 88.3% (1130 of 1279) of clones that bound to the original SARS-CoV-2 RBD also bound to at least one of six variants. Many of these phage clones were able to bind to several SARS-CoV-2 virus variants;

for example, 386 phage clones bound to all five of the single amino acid variants and 147 clones bound to all five single amino variants plus the RBD Gamma variant (Fig. 5a). We were able to validate the differential binding predicted using SLISY by comparing it to ELISA-based testing using phage previously derived from binding to the original SARS-CoV-2 (Fig. 5b, c). This analysis revealed perfect concordance between SLISY and the ELISA. Four other variants of interest were available in either the S1 (D614G and Beta - B.1.351) or FL (SARS-CoV-1 and Alpha - B.1.1.7) formats. Similar to above, there was perfect concordance between SLISY and the ELISA based test for both S1 and FL (Supplementary Fig. 5).

While some of our full-length antibody clones did bind to the variants, none were able to simultaneously neutralize the Beta, Gamma or Delta variants, three viral variants that were highly clinically relevant (Supplementary Data 5 and 6). Therefore, we used SLISY from existing biopannings to quickly identify new clones against these three variants as well as the original strain (Fig. 6a).

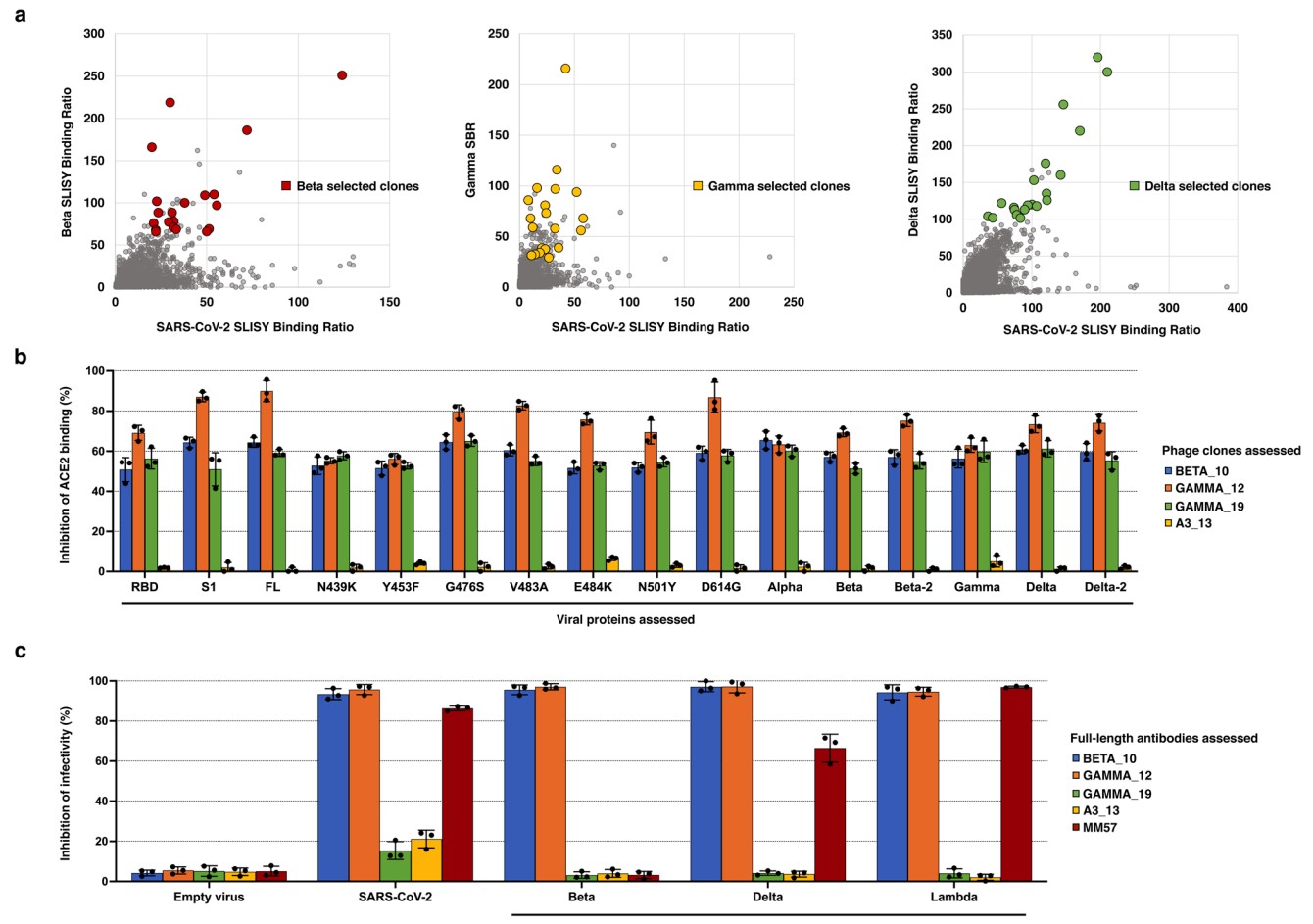

**Fig. 6 | Using SLISY to identify clones that binding across multiple variants.**
**a** Selecting variant binding clones for validation. After applying the enriched pool of scFvs from biopanning with the original SARS-CoV-2 to variant proteins, the original SARS-CoV-2 SBR for each clone was plotted against a variant SBR. Clones were selected for testing based on high SBRs for both the original and variants. Twenty clones were selected from each of the original-variant pairings. **b** Broad spectrum neutralization by superclone scFvs Beta_10, Gamma_12, and Gamma_19. The full-length scFv phage clones were applied to wells pre-coated with variant SARS-CoV-2 spike proteins, followed by the addition of recombinant ACE2-His protein. Negative control is A3-Clone 13 scFv phage. **c** Ability of full-length antibodies to block variant pseudovirus infectivity. Converted full-length antibodies were incubated with pseudovirus expressing variant SARS-CoV-2 spike protein and ACE2-expressing HEK-293T cells for 48 h. Infectivity was measured by luciferase activity. Error bars represent standard deviation of means ($n = 3$). Source data are provided as a Source Data file.

Because clones with the most potent neutralizing capacity do not necessarily have the highest SBR, we selected twenty candidate clones against each of the three variants. All tested clones demonstrated strong binding to the original strain and its corresponding variant, with many cross-reactive to multiple variants (Supplementary Data 7). In a competitive ELISA format, we also observed that at least 2 out of the 20 clones from each set demonstrated significant inhibition greater than 50% against its variant (Supplementary Data 8). With this, we quickly identified clones that were able to block ACE2 binding to multiple variants. In particular, phage clones Beta_10, Gamma_12, and Gamma_19 simultaneously inhibited more than 50% of ACE2 binding to the original, Alpha, Beta, Gamma, and Delta variants. (Fig. 6b). When converted to full-length antibodies, Beta_10 and Gamma_12 were able to block infectivity of original SARS-CoV-2 spike pseudovirus as well as variant pseudoviruses of Beta, Delta, and the more recently emerged Lambda (Fig. 6c). By selecting clones using multiple comparisons with SLISY, we rapidly and successfully identified several broad spectrum superclones that warrant further evaluation.

In summary, we were able to rapidly identify phage clones that could selectively bind to all 14 clinically relevant SARS-CoV-2 viral strains tested and three superclones that could each bind to every variant. We expect that SLISY could be applied to many types of display libraries and used to develop diagnostic and therapeutic proteins, including antibodies, against a variety of important biomedical targets.

## Methods

### Phage library construction and expansion
Methods for scFv phage display library construction and expansion were described previously[28]. Briefly, SS320 bacteria (Lucigen, 60512-2) grown to mid-log phase were supplemented with 2% glucose and infected with M13K07 Helper phage (Antibody Design Labs, PH010L) and library phage. After 1 h, cells were resuspended in 2xYT medium with carbenicillin (100 μg/mL), kanamycin (50 μg/mL), and IPTG (50 μM) and grown overnight at 30 °C. The following morning, the bacterial culture was pelleted at high speed (12,000 g) to obtain clarified supernatant. The phage-laden supernatant was precipitated on ice for 40 min with a 20% PEG-8000/2.5 M NaCl solution at a 1:4 ratio of PEG/NaCl:supernatent. After precipitation, phage was centrifuged at 12,000 g for 40 min and resuspended in 1X TBS, 2 mM EDTA. Phage from multiple tubes were pooled, re-precipitated, and resuspended to

an average titer of $1 \times 10^{13}$ cfu/mL. The precipitated phage represents ~250-fold coverage of the library.

### Selection for phage binding to HLA-A3 on CFPAC parental cells

For round 1 of cell-based panning, $100 \times 10^6$ CFPAC parental cells (ATCC, CRL-1918) were harvested and washed in PBS. The cells were directly resuspended in the pooled phage library and incubated at 4 °C for 1 h with rotation. For the A3 clone spike-in condition, phage clone A3-Clone 20 was initially added to the pooled library for a final amount of 1:100000 phage. After pelleting, the cells were washed 5X in PBS. To recover the phage for reinfection, 1 mL of Elution Buffer (0.2 M glycine at pH 2.2) was used to resuspend cells for 10 min at 20 °C. The solution was neutralized with 150 μL of Neutralization Buffer (1 M Tris HCl at pH 9.0). The neutralized phage were then used to infect a 10 mL SS320 culture at mid-log phase with the addition of M13K07 helper phage (MOI of 4) and 2% glucose. As previously described above, bacteria were grown overnight and phage particles in conditioned medium were precipitated with PEG/NaCl. Starting in round 2 and for the remainder of biopanning, $15 \times 10^6$ CFPAC HLA A3 knockout cells (custom in-house line generated from parental) were washed in PBS and used for negative biopanning. Like positive biopanning, the phage was incubated for 1 h at 4 °C with rotation before cells were pelleted and the supernatant containing unbound phage was used for positive biopanning. The amount of CFPAC parental cells used for positive biopanning decreased with each round: $15 \times 10^6$ cells for round 2, $10 \times 10^6$ cells for round 3, and $5 \times 10^6$ cells for round 4. At the end of each round, cells were pelleted and phage particles were eluted for infection of overnight SS320 cultures.

### Biopanning for phage binding to SARS-CoV-2

His-tagged SARS-CoV-2 and MERS antigens (RBD, S1, and FL) were conjugated to HisPur™ Ni-NTA magnetic beads (Thermo Fisher, 88832) at a concentration of 16 μg protein/mg beads following manufacturer's protocol (Supplementary Data 9). In order to remove any phage recognizing a portion of the magnetic beads itself, the precipitated library was applied to naked 500 μL washed Ni-NTA magnetic beads overnight at 4 °C with rotation. After negative biopanning, the supernatant containing unbound phage was applied to beads conjugated with 10 μg of each SARS-CoV-2 protein (RBD, S1, FL) for a total of 30 μg protein for 1 h at 4 °C. The beads were washed 3X with TBST (TBS + 0.05% Tween-20) with the last wash overnight at 4 °C. Phage were eluted, neutralized, and used to infect bacteria as previously described above. Beginning in round 2, we separated the precipitated round 1 phage into three different biopanning strategies corresponding to RBD, S1, or FL. We applied an overnight negative biopanning of 30 μg of the corresponding MERS protein-conjugated beads (i.e. for SARS-CoV-2 RBD panning, use MERS RBD) prior to positive biopanning. This remained constant for rounds 3 and 4. However, the amount of SARS-CoV-2 protein used for positive biopanning decreased from 4 μg in round 2 to 1 μg in round 3 to 0.5 μg in round 4. After each negative biopanning, the sample was applied to a magnet to isolate the supernatant for positive biopanning. After each round, phage were eluted and used to infect bacteria for overnight cultures.

### Sanger sequencing of full-length scFv phage clones to validate constructs

To identify each clone by Sanger sequencing, 1 μL of bacteria was mixed following manufacturer's protocol with Q5 Hot Start High-Fidelity 2X Master Mix (New England Biolabs, M0494X) and forward and reverse primers (phiS2: 5'-ATGAAATACCTATTGCCTACGG and psiR2: 5'-CGTTAGTAAATGAATTTTCTGTATGAGG). The cycling conditions used are as described:

| 1 cycle | 1 min at 98 °C |
|---|---|
| 3 cycles | 10 s at 98 °C, 10 s at 70 °C, 15 s at 72 °C |
| 3 cycles | 10 s at 98 °C, 10 s at 67 °C, 15 s at 72 °C |
| 3 cycles | 10 s at 98 °C, 10 s at 64 °C, 15 s at 72 °C |
| 3 cycles | 10 s at 98 °C, 10 s at 61 °C, 15 s at 72 °C |
| 20–30 cycles | 10 s at 98 °C, 10 s at 58 °C, 15 s at 72 °C |
| 1 cycle | 5 min at 72 °C |
| | Hold at 10 °C |

PCR products were submitted to Genewiz (South Plainfield, NJ) for Sanger Sequencing. Entire sequence of scFv was identified using SnapGene (San Diego, CA).

### Sequencing-Linked ImmunoSorbent assaY (SLISY)

**For cell-based SLISY.** For each cell line, CFPAC parental and CFPAC A3 KO, $10 \times 10^6$ cells were trypsinized and washed 3X with PBS. Cell pellets were resuspended in PBS at a concentration of $10 \times 10^6$ cells/mL. Next, 10 μL of polyclonal phage ($10^{13}$ titer) was applied to both samples and incubated for 1 h at 20 °C with continuous rotation. Samples were pelleted and washed 3X with PBS. After washing, the samples were resuspended in QuickExtract™ DNA Extraction Solution (Lucigen, QE09050) according to manufacturer's protocol. Samples were passed through the QIAshredder (Qiagen, 79656) and used directly for PCR amplification of CDRH3 region.

**For SARS-CoV-2 SLISY.** His Tag antibody plates (Genscript, L00440C) were individually coated with the spike proteins desired for SLISY comparisons. The spike proteins were diluted in PBS to a concentration of 0.5 μg/mL and incubated at 4 °C for 12 h. Plates were vigorously washed 6X times with 1X TBST. Next, 100 μL of polyclonal phage ($10^{13}$ titer) was applied to each of the wells for comparison and incubated for 1 h at 4 °C. Next, the plates were washed again 6X times with TBST. After washing, 20 μL of Elution Buffer (0.2 M glycine at pH 2.2) was applied to wells for 20 min at 20 °C. Without washing, 3 μL of Neutralization Buffer (1 M Tris HCl at pH 9.0) was directly added to wells and total volume in well (~23 μL) was recovered for PCR amplification.

**PCR amplification.** Eluted phage was amplified using the following primers (Forward: GGATACCGCTGTCTACTACTGTAGCCG, Reverse: CTGCTCACCGTCACCAATGTGCC) which flank the CDR-H3 region. The sequences at the 5'-ends of these primers incorporated molecular barcodes to facilitate unambiguous enumeration of distinct phage sequences. The protocols for PCR-amplification and sequencing are described in Kinde et al. (2011)[27]. Sequences were demultiplexed and processed to extract and translate CDRH3 regions using custom python software (https://doi.org/10.5281/zenodo.7154344) and analyzed using SQL databases (MSSQL) and Microsoft Excel. The SLISY Binding Ratio was calculated by comparing the phage eluted from the target well/cells to the phage eluted from the non-target (negative) well/cells.

$$\text{SLISY Binding Ratio (SBR)} = \frac{\text{Clone UID in Target Pool}}{\text{Clone UID in Negative Pool}}$$

Clones that have a UID of zero are given a value of 0.5. Because wells were loaded with equivalent amounts of phage and we were using molecular barcodes to count molecules, we did not normalize for total reads. For comparison, we also calculated a Panning Enrichment Ratio (PER) by dividing the fraction of the phage expressing the specific CDR-H3 in the population after growth of the bound phage by the fraction present in the input material and then converting the value to log base 2.

## Long-read sequencing of candidate clones selected by SLISY

SLISY initially uses sequencing of just the CDR-H3 region of the scFv to determine whether *any* promising clones exist in the library following biopanning. If so, we used more extensive sequencing to determine the sequence of the entire scFv. In order to obtain the complete variable regions of the heavy and light chains of the scFv, long-read sequencing utilized a custom protocol with three reads (Supplementary Fig. 1). To efficiently multiplex samples, 96 forward primers were designed with unique well barcodes to serve as sample indexes and a Unique IDentifier (UID) of 14 random nucleotides to serve as molecular barcodes (Supplementary Data 10). Thus, each primer contained both a well barcode and the means to identify up to $4^{14}$

unique molecules per well. Each 25 μl reaction consisted of 12.5 μl of Q5 High-Fidelity 2X Master Mix (New England Biolabs, Ipswich, MA, cat #0491 L), 1 μl of 10 μM forward primer (IDT, Coralville, IA), 1 μl of 10 μM reverse primer (IDT), 1 μl of phage, and 9 μl nuclease-free water. Amplification conditions were as follows: 1 denaturation cycle of 98 °C for 3 min; 10 amplification cycles of 98 °C for 10 s, 61 °C for 2 min, and 72 °C for 2 min; followed by an infinite hold at 4 °C. Amplicons were then sequenced on a MiSeq using the following custom primers designed to sequence all variable regions of the scFv: read 1 primer – 5′-ACTGGCCGTCGTTTTA CGTCG-3′, read 2 primer – 5′-TGCGCTAATGGTAAAGCGACCTTT CACGCTAT-3′, and index primer – 5′-CTGCAGATGAATAGTCTGCGT GCAGAGGATACAGC-3′. Run parameters were as follows: read 1 – 220 cycles, read 2 – 135 cycles, and index – 145 cycles. Reads passing Illumina chastity filters were included in subsequent analysis.

## Cloning and expression of full-length candidate scFvs identified by SLISY

Based on the long-read sequencing, geneblocks corresponding to the entire scFv were ordered from Integrated DNA Technologies and resuspended at a concentration of 0.05 μM in TE Buffer. The pADL-10b phagemid vector was digested using BglI enzyme (New England Biolabs, R0143S) with Antarctic Phosphatase (New England Biolabs, M0289L) in manufacturer recommended Buffer 3.1 for 12 h at 37 °C. Following PCR purification (Qiagen, 28104), 100 ng of linearized pADL-10b plasmid and 1 μL of each resuspended geneblock was added to NEBuilder® HiFi DNA Assembly Master Mix (New England Biolabs, E2621X) for a reaction total volume of 20 μL that was incubated at 50 °C for 1 h. One μL of the ligation product was mixed with 25 μL of electrocompetent SS320 cells. This mixture was electroporated using a Gene Pulser electroporation system (Bio-Rad, BZA648860) and allowed to recover in Recovery Media (Lucigen, 80026-1) for 1 h at 37 °C with shaking. Next, 100 μL of 1:500 dilution of transformed cells were plated on 2xYT agar plates supplemented with carbenicillin (100 μg/mL) and 2% glucose. Cells were grown at 37 °C for 12 h. Individual colonies were inoculated into 200 μl of 2xYT medium containing 100 μg/mL carbenicillin and 2% glucose and grown for three hours at 37 °C. The cells were then infected with $1.6 \times 10^7$ M13K07 helper phage and incubated for an additional 1 h at 37 °C. The cells were pelleted, resuspended in 300 μL of 2xYT medium containing carbenicillin (100 μg/mL) and kanamycin (50 μg/mL), and grown overnight at 30 °C. Clones were submitted for Sanger sequencing and inserts were confirmed using SnapGene. Once verified, cells were pelleted and 100 μL of phage-laden supernatant was used to infect 9 mL of bacteria to produce phage overnight as described above.

## Flow cytometry of cell lines to validate HLA-A3-binding clones

CFPAC parental and HLA-A3 KO lines were harvested, washed with PBS, and resuspended in ice-cold flow cytometry staining buffer (PBS, 0.5% BSA, 2 mM, EDTA, 0.1% sodium azide) at a concentration of $10 \times 10^6$ cells/mL. Next, 10 μL of precipitated phage ($10^{13}$ titer) was applied to 100 μL of both cells and incubated on ice for 15 min. After

washing 3X with staining buffer, cell pellets were resuspended to same concentration ($10 \times 10^6$ cells/mL) and stained with rabbit anti-M13 polyclonal antibody (Novus Biologicals, Littleton, CO) at 1:100 final dilution on ice for 15 min. After washing 3X again, cell pellets were resuspended and stained with PE donkey anti-rabbit IgG antibody (Biolegend, San Diego, CA) at 1:100 final dilution for 15 min on ice. After a final wash 3X, stained CFPAC cells were analyzed using a LSRII flow cytometer (Becton Dickinson, Franklin Lakes, NJ) to measure mean fluorescence intensity. For gating, any PE signal above background in unstained cells was considered positive (Supplementary Fig. 6).

## ELISAs for binding and specificity

His Tag antibody plates were coated with a 0.5 μg/mL solution of spike protein antigens diluted in PBS at 4 °C for 12 h and washed with 1X TBST.

**For phage.** To test binding and specificity of phage to spike antigens, precipitated phage ($10^{13}$ titer) was diluted 1:1000 in PBS and 100 μL was added to each well and incubated for 1 h at 4 °C. After vigorous washing, the bound phage were then incubated with 100 μL of rabbit anti-M13 antibody diluted 1:5000 in 1X TBST for 1 h at 4 °C. Following 6 washes with 1X TBST, wells were then incubated with 100 μL of goat anti-rabbit IgG (H + L) antibody HRP (Thermo Fisher, A27036) diluted 1:10,000 in 1X TBST for 1 h at 4 °C.

**For converted antibodies.** Each converted full-length IgG antibody clone was diluted to 1.0 ug/mL in PBS and 100 μL was added to each well with 1 h incubation at 4 °C. Wells were then incubated directly with 100 μL of goat anti-human IgG Fc-HRP antibody (Abcam, ab98624) diluted 1:10,000 in 1X TBST for 1 h at 4 °C. After 6 more 1X TBST washes, 100 μL of TMB substrate (Biolegend, 421101) was added and allowed to develop. The reaction was quenched with 50 μL of 2 N sulfuric acid. Absorbance at 450 nm and 540 nm was measured with a Synergy H1 Multi-Mode Reader (BioTek, Winooski, VT). O.D. readings were measured as absorbance at 450 nm minus that at 540 nm.

## Competitive ELISAs to identify phage and mAbs that inhibit ACE2 binding

His Tag antibody plates were coated with a 0.5 μg/mL solution of spike protein antigens diluted in PBS at 4 °C for 12 h then washed with 1X TBST. To test blocking of ACE2 binding, 100 μL of precipitated phage ($10^{13}$ titer) or 10 μg/mL of converted antibody was applied to each well and incubated for 1 h at 4 °C. After vigorous washing, the wells were then incubated with 100 μL of recombinant ACE2-His protein (RayBiotech Life, Inc, Peachtree Corners, GA) for 1 h at 4 °C. After 6X washing, 100 μL of rabbit anti-6X His tag polyclonal antibody (Abcam, ab9108) diluted 1:1,000 in PBS was added and incubated for 1 h at 4 °C. Following 6 washes with 1X TBST, wells were then incubated with 100 μL of goat anti-rabbit IgG (H + L) antibody HRP diluted 1:10,000 in 1X TBST for 1 h at 4 °C. Competitive ELISAs were developed and measured identical to binding ELISAs described above. A negative control antibody, SARS-CoV-2 spike mouse mAb (40591-MM42) (Sino Biological, Wayne, PA) that binds to the SARS-CoV-2 spike protein but does not neutralize, and a positive control antibody, SARS-CoV-2 spike mouse mAb (40592-MM57) (Sino Biological) that blocks interaction with ACE2 were used at 10 μg/mL. An ACE2-His alone condition without phage or antibody served as the baseline signal for spike-ACE2 binding. Blocking potential was measured as a decrease in signal from ACE2-His alone.

## cPASS: SARS-CoV-2 surrogate virus neutralization test (sVNT)

Assays were performed following manufacturer's protocols for measuring neutralization using the SARS-CoV-2 sVNT kit (Genscript)[31]. Briefly, 100 μL of 1:1 and 1:10 phage ($10^{13}$ titer) diluted in Sample

Dilution Buffer was mixed with diluted HRP-RBD solution with a volume ratio of 1:1. Mixtures were incubated at 37 °C for 30 min. For antibodies, 100 μL of 1 μg/mL and 0.1 μg/mL antibody diluted in Sample Dilution Buffer was mixed with HRP-RBD solution. Provided positive and negative controls were prepared similarly following manufacturer protocols. Next, 100 μL of mixtures were applied to wells pre-coated with recombinant ACE2 protein and incubated at 37 °C for 15 min. After washing plates with 260 μL 1X Wash Solution four times, 100 μL of provided TMB solution was added and the reaction developed for 15 min at 25 °C. Reaction was quenched with Stop Solution and plate read at 450 nm and 540 nm absorbance. According to manufacturer's protocol, neutralization was measured as a decrease in signal greater than or equal to 30% of signal from the negative control well.

$$Inhibition = \left(1 - \frac{\text{OD value of Sample}}{\text{OD value of Negative Control}}\right) \times 100\%$$

### Production and purification of SARS-CoV-2 specific antibodies

For IgG Antibodies, the light and heavy chain variable sequences of the scFvs were grafted on the chains of 4D5/trastuzumab and cloned into pcDNA3.4 backbone with a mouse IgKVIII leader signal peptide as we have done before in Hsiue et al. (2021)[24]. Freestyle 293-F cells were transfected with light and heavy chain plasmids at a ratio of 1:1 using PEI at 1:3 at a concentration of $2 \times 10^6$ to $2.5 \times 10^6$ per ml and incubated for 6 days at 37 °C. Transfection and expression of the antibodies were carried out at the Eukaryotic Tissue Culture Core Facility at Johns Hopkins University. The media was harvested by centrifugation, filtered through a 0.22 μm PES membrane and purified via protein A affinity chromatography on a HiTrap MabSelect SuRe column (Cytiva, Malborough, MA) with the following running buffer: 20 mM Sodium Phosphate, 150 mM NaCl pH 7.2. The antibodies were eluted with a linear gradient from 0 to 100 mM glycine pH 3.0 over 30 column volumes. Fractions were collected in prefilled tubes with 1 M TRIS pH 9.0. The fractions were quantified by SDS-PAGE gel electrophoresis and the fractions of pure antibody were pooled and dialyzed into 20 mM Sodium Phosphate, 150 mM NaCl pH 7.2. Gel filtration chromatography was used for further purification in 20 mM Sodium Phosphate, 150 mM NaCl pH 7.2 with the Superdex 200 increase, 10/300 GL column (Cytiva). Final fractions were quantified by SDS-PAGE gel electrophoresis and the fractions containing antibody were frozen with liquid nitrogen and stored at −80 °C.

### Surface plasmon resonance (SPR) affinity measurements

SPR experiments were carried out on a Biacore T200 (Cytiva) at 25 °C of a CM5 chip. Protein A/G was diluted (1:25 dilution, 1 μM diluted concentration) in 10 mM sodium acetate buffer at pH 4.5 and immobilized on all flow cells (Fcs) of the CM5 chip to a level of ~4100 response units (RU) using standard amine coupling chemistry. HBS-P (10 mM Hepes pH 7.4, 150 mM NaCl, 0.05% v/v surfactant P20) was used as the immobilization and capture running buffer. Approximately ~70-200 RU of each SLISY-selected antibody was captured onto Fcs 2 through 4. Fc1 was used as reference subtraction. Single-cycle kinetics were performed for the analytes binding to the captured ligands in the presence of HBS-P by increasing concentrations (5, 20, 80, 160, and 320 nM, four-fold dilutions) of purified target analytes flowed over Fc 1-4 at a rate of 50 μL/min. The analytes used were SARS-CoV-2 RBD, SARS-CoV-2 S1, and SARS-CoV-2 FL (Supplementary Data 9). The contact and dissociation times were 120 s and 600 s, respectively. One 20 s injection of Glycine pH 2.0 was used for surface regeneration. This regeneration also took away captured ligands. Therefore, ligands were captured in the beginning of every cycle. Binding responses for kinetic analyses were reference and blank subtracted. All curves were fit with a 1:1 kinetic binding model using Biacore Insight evaluation software. All SPR measurements were done in triplicates.

### SARS-CoV-2 Spike pseudotyped lentiviral assay for neutralization

SARS-CoV-2 S-protein pseudotyped replication incompetent lentiviral particles were produced by first transfecting HEK-293T (ATCC, CRL-3216) with GeneJuice transfection reagent (Millipore-Sigma) and SARS-Related Coronavirus 2, Wuhan-Hu-1 Spike-Pseudotyped Lentiviral Kit (Spike-Pseudotyped Lentiviral Kit, NR-52948; BEI Resources Repository, Manassas, VA). Viral supernatant was collected 48 h after transfection and filtered through a 0.45 μM filter. For variant viruses, commercially available pseudotyped luciferase rSARS-CoV-2 spike virus for the Beta, Delta, and Lambda strains were obtained from Creative Biosciences (Shirley, NY). $1.25 \times 10^4$ 293 T cells engineered to express hACE2 receptor (hACE2.293 T cells) were plated on day −1 in black 96-well microplates (Corning, Corning, NY)[37]. Parental 293 T cells served as a control for nonspecific cell transduction. On Day 0, S-pseudoviral particles +/− the indicated antibodies were added and the plate centrifuged at 800 g for 30' at 32 C. Optimal amounts of S-pseudoviral particles for each virus were titrated up to maximal signal-to-noise without antibodies. Cells were then incubated at 37 C in 5% $CO_2$ for 48 h at which time the viral-containing supernatant was aspirated and fresh media containing 150 ug/ml D-Luciferin (Millipore-Sigma) for the original strain or *Renilla* Luciferase (Promega, Madison, WI) for variant viruses was added. BLI was measured and reactive light units (RLU) determined after subtraction of virus-only background. Inhibition of infectivity by each antibody was calculated as percentage decrease in BLI over baseline (virus without antibodies). The assay was performed in experimental triplicate.

### Epitope binning of SARS-CoV-2 mAbs

An indirect ELISA format was used to identify whether identified neutralizing antibody clones could compete for the same antigenic epitope. Briefly, antibodies were biotinylated and an initial indirect ELISA was performed to calibrate the appropriate concentration (OD value approaches 1.5) of each antibody for the binning assay against the SARS-CoV-2 RBD-His. Antibodies with low binding activity on the RBD-His were excluded from downstream assays due to difficulty in showing self-competition. Biotinylated antibody is mixed with each free antibody with the calibrated concentration through a volume ratio of 1:1 and applied to an indirect ELISA format to analyze competition within each pair. Antibody pairs are categorized as likely having the same epitope if one of the competition values ($OD_{mAb1-mAb2}$) is lower than the self-competition value of the other antibody ($OD_{mAb2}$) and the other competition value ($OD_{mAb2-mAb1}$) demonstrates an inhibition of greater than 30% than that of the biotinylated antibody alone ($OD_{mAb1}$). Epitope binning assay performed and analyzed by Genscript.

$$Inhibition_{mAb1-mAb2} = \left(1 - \frac{OD_{mAb1-mAb2} - OD_{mAb2}}{OD_{mAb1} - OD_{mAb2}}\right) \times 100\%$$

### Statistics & reproducibility

Unless otherwise indicated, error bars represent the standard deviation of three biological replicates. No statistical method was used to predetermine sample size. Experiments were performed in replicates of three to generate a standard deviation of means. No data were excluded from the analyses.

### Reporting summary

Further information on research design is available in the Nature Portfolio Reporting Summary linked to this article.

## Data availability

The sequencing data generated in this study have been deposited in the European Nucleotide Archive (ENA) at EMBL-EBI under accession number "PRJEB58033" and are limited to non-commercial and research use per Johns Hopkins legal requirements unless explicitly granted by

institution. There is a sample dataset of a MiSeq run of a SLISY experiment for testing that is also available at Zenodo (https://doi.org/10.5281/zenodo.7154344). Source data are provided with this paper.

## Code availability

The code used in the manuscript to process sequences and translate CDRH3 regions is a custom python software available at Zenodo (https://doi.org/10.5281/zenodo.7154344). The SLISY SQL databases (MSSQL) are available from the corresponding author (Kenneth Kinzler: kinzlke@jhmi.edu) and are limited to non-commercial and research use per Johns Hopkins legal requirements unless explicitly granted by institution. Requests for the databases will be fulfilled within one week.

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

## Acknowledgements

The expression of antibodies was carried out at the Eukaryotic Tissue Culture Facility of The Johns Hopkins University School of Medicine. We thank Janine Ptak, Lisa Dobbyn, Natalie Silliman and Joy Schaefer for excellent support of Next Generation sequencing. Funding for this work was supported by The Virginia and D.K. Ludwig Fund for Cancer Research, The Bloomberg-Kimmel Institute for Cancer Immunotherapy, and NIH Cancer Center Support Grant P30 CA006973.

## Author contributions

S.L., A.K.M., N.P., B.V., and K.K. conceived the original project. S.L. and A.K.M. led the experimental work with phage biopanning, the development of SLISY, and the generation of phage clones with supervision from S.S., S.Z., N.P., B.V., and K.K. The full-length antibody purification and characterization work was led by P.A.A., K.M.W., Z.C., and Y.L. with supervision from S.B.G. The sequencing design and analysis of SLISY was led by S.L., A.K.M., and M.P. under the supervision of N.P., B.V., and K.K. The validation of specificity, infectivity, and neutralization assays were performed by S.L., A.K.M., and I.C. with supervision from C.L.B., C.B., S.Z., N.P., B.V., and K.K. All authors interpreted the data and wrote the manuscript.

## Competing interests

C.L.B. and I.C. have submitted a patent application to the US PTO pertaining to the binding of the SARS-CoV-2 vial envelope by engineered NK cells (application number PCT/US2021/053042). C.L.B. has received research funding from Merck, Sharp, and Dohme, Inc., Bristol Myers Squibb, and Kiadis Pharma. B.V., K.W.K., and N.P. are founders of Thrive Earlier Detection. K.W.K. and N.P. are consultants to and were on the Board of Directors of Thrive Earlier Detection. B.V., K.W.K., N.P., and S.Z. own equity in Exact Sciences. B.V., K.W.K., N.P., and S.Z. are founders of, hold or may hold equity in, and serve or may serve as consultants to ManaT Bio. B.V., K.W.K., N.P., and S.Z. are founders of, hold equity in, and serve as consultants to Personal Genome Diagnostics. S.Z. has a research agreement with BioMed Valley Discoveries. S.B.G. is a founder and holds equity in AMS LLC. S.B.G is a consultant to Genesis. K.W.K. and B.V. are consultants to Sysmex, Eisai, and CAGE Pharma and hold equity in CAGE Pharma. B.V. is also a consultant to Catalio. K.W.K., B.V., S.Z., and N.P. are consultants to and hold equity in NeoPhore. N.P. is an advisor to and holds equity in CAGE Pharma. C.B. is a consultant to Depuy-Synthes and Bionaut Pharmaceuticals. The companies named above, as well as other companies, have licensed previously described technologies related to this paper from Johns Hopkins University. B.V., K.W.K., S.Z., N.P., and C.B. are inventors on some of these technologies. Licenses to these technologies are or will be associated with equity or royalty payments to the inventors as well as to Johns Hopkins University. The terms of all these arrangements are being managed by Johns Hopkins University in accordance with its conflict of interest policies. The remaining authors declare no competing interests.
