## [Peer Review File · Nature Communications]

REVIEWER COMMENTS

Reviewer #1 (Remarks to the Author):

In this manuscript, Steve Lu et al introduced SLISY (Sequencing-Linked ImmunoSorbent Assay), which allows the high throughput selection of antigen binders from the single chain variable fragment (scFv) displayed phage library.

The basic procedures of this platform are as follows. First, high throughput NGS analysis of HCDR3 sequence after a round of biopanning on spike protein or its fragment of SARS-CoV2 and MERS was performed. From the analysis on the enrichment pattern of scFv clones based on HCDR3 sequence, the clones specifically enriched on the antigens were selected. Then to extract the whole scFv sequences, a long read sequencing covering over 600 nucleotide residues were performed. This sequence information was used to synthesize the immunoglobulin genes for further characterization.

Major concerns

1. It is very difficult for readers to understand the logic how the whole scFv sequence could be extracted from the long read sequencing. The nature of scFv libraries used in this study was not described in this manuscript. And the extended Data Fig. 8 referred by authors as "describing the NGS protocol (Line 355)" is actually an irrelevant one. Two scFv libraries in reference 23 and 31 (Line 249) were referred to be used in this study but actually described in detail in reference 38 (PNAS 2015 112 p9967) of the reference 31 (J Biol Chem 2019 294 p19322). Based on the description of this reference, both scFv libraries are based on the humanized 4D5 (trastuzumab) framework. The first library varies CDRs L3, H1, H2, and H3 and describe in detail in the reference 38 (PNAS 2015 112 p9967) of reference 28 (J Biol Chem 2019 294 p19322). The second library was described to be varied in CDR L2 in addition to the other four CDRs and uses TRIM (trinucleotide mutagenesis) technology to generate a greater degree of diversity. But the nature of TRIM technology is not described in this manuscript or in reference 28. The extended Data Figure 1 is not providing information unsatisfactory to figure out how this long read sequencing is achieved. It can be speculated that the all the scFv genes in these scFv libraries share the same framework of humanized 4D5 and differ only in LCDR (2) and 3 and all HCDRs. If it is the case, long read sequencing can be achieved by primers based on framework regions. If this is the case, this strategy can be applied to only to synthetic scFv library constructed on the identical framework and cannot be used in other naïve, synthetic or immune scFv libraries with diverse panel of frameworks. This significantly limit the value of SLISY platform described in this manuscript. The authors are advised to clarify this point,
2. The authors used term "neutralize" in this manuscript. But the term "neutralizing activity of mAb" should be defined in in vitro or in vivo assay using SARS-CoV2 virus or at least pseudovirus. The authors actually employed competition ELISA.

Reviewer #2 (Remarks to the Author):

Discovering recombinant antibodies is about to change thanks to ease of access to sequence data. But how to use that to actually get better antibodies? There is an apparent need to un-blackbox what is happening during selection and what is being enriched between multiple rounds of selection. The authors present a version of phage-ELISA coupled with NGS to link binding phenotype to sequence information that attempts to identify rare cross-reactive binders that are easily missed or might even get lost across multiple rounds of phage-display selection. This approach appears to tackle the often encountered "growth bias" quite nicely by sequencing the whole phage output and

calculating a ratio of sequences encountered for the specific antigen against a control. It appears to be easily adaptable to other antigens and selection workflows but the authors should provide some more background on the phage library (such as diversity and the scaffold used). I have little doubt NGS will become much more widely adapted in phage-display panning campaigns and this study could serve as a template in how to use sequence information to prioritize candidate clones.

General comment: Occasionally the language seems sloppy, with interpretation of results appearing to be over-generalized from what is supported by the data. For example the anti-HLA phage display campaign could benefit from being treated more quantitatively (see specific comments below). It is unclear to me which main figure each of the extended data figures is related to (for example: Extended Data Figures 1, 2, 3, 4, 5, 6, 7 Extended Data Table 1, 2 all get referenced between Figure 1a and 1b) and the usage of figure captions instead as a supplementary results section (ex. Extended Data Figure 3) makes the manuscript occasionally very tedious to read.

L. 88-89 "universally amplify the CDR-H3 region of any scFv" - do those primers really work with any scFv from any library or what is the range of the "any" here?

L. 95-96 "full length sequence of any phage" - not sure what that means. Do the authors mean "phagemid"? How long is the sequence of a phage?

L.98-99 "these methods (...) are described in detail in Materials and Methods" - seems to be rather unnecessary. I would assume that is where they would be described.

L. 104-105 "(...) desirable clones, which would be difficult to identify using a traditional growth based screening approach." - please elaborate. It is not obvious what is considered desirable and "difficult to identify" is extremely hand-wavy. I have no doubt there are interesting things to be found, but this seems like the authors are guessing rather than measuring.

L. 109-110 "(...) improve both the number and quality of clones (...)" - again, considering this is the combined results/discussion section I am missing some sort of descriptive presentation of the obtained data. What is the "quality", what is being improved there and by how much? "Improving the number" probably means more? How much more?

L.118-119 "Four rounds of traditional biopanning (...)" - Please be specific about what you mean with "traditional".

L. 125 "scFv clones with SBR > 10" - Why 10? Please explain why you choose this specific value as a cutoff.

L. 149 "a type of digital ELISA" - please explain further. What makes this assay digital?

L. 178ff "Converting scFvs to full-length antibodies (...) can be problematic. We addressed this issue by converting the scFv regions (...) into full-length IgG antibodies." - I would consider it helpful if the authors could provide some detail into how conversion into IgG addresses the issues of conversion into IgG. What does "conversion" imply in this setting? Grafting CDR regions, cloning VH and VL?

L. 207 & L. 213 "ELISAs for millions of clones" vs 1279 clones - I am confused. Did the authors test millions of clones using SLISY or did they test 1279? Could you not do ELISAs for the 1279 clones? Or were those derived from millions of clones that went into the assay?

L. 248 "Methods for scFv phage display library construction" - please provide a reference for the construction of the library used in this study. Ref 23 and Ref 31 refer themselves to another study (the same in this case). If the authors used the library from the Skora et al. paper, why not reference that? It is rather tiresome having to dig through another level of references within references.

Reviewer #3 (Remarks to the Author):

In. their manuscript ("SLISY: Rapid identification of antibodies to evolving viral pathogens") the authors develop a novel strategy to identify binders by phage display by using differential enrichment after panning, in combination with high throughput sequencing. This is an interesting and innovative strategy, in principle suitable for publication in Nature Communications, and I greatly enjoyed reading the manuscript.

However, in its current form, I have two concerns with the paper:

1) The approach in many respects is reminiscent of the combinatorial scanning approach on phage developed at Genentech over 20 years ago (by Greg Weiss, Sachdev Sidhu and others). See: <https://www.pnas.org/doi/10.1073/pnas.160252097> and other publications. This needs to be extensively discussed and placed into historical context

2) More importantly, the characterisation of the selected antibodies is limited to ELISA type assays. For the SARS-CoV-2 field in particular a more extensive characterisation using at least pseudoparticle assays, and preferably structural and vivo characterisation (human ACE-2 mice or hamster) would be expected. The assays outlined here would, in particular, not detect neutralising antibodies to important class 3 and 5 (non-ACE2) epitopes.

REVIEWER COMMENTS

Reviewer #1 (Remarks to the Author):

In this manuscript, Steve Lu et al introduced SLISY (Sequencing-Linked ImmunoSorbent AssaY), which allows the high throughput selection of antigen binders from the single chain variable fragment (scFv) displayed phage library.

The basic procedures of this platform are as follows. First, high throughput NGS analysis of HCDR3 sequence after a round of biopanning on spike protein or its fragment of SARS-CoV2 and MERS was performed. From the analysis on the enrichment pattern of scFv clones based on HCDR3 sequence, the clones specifically enriched on the antigens were selected. Then to extract the whole scFv sequences, a long read sequencing covering over 600 nucleotide residues were performed. This sequence information was used to synthesize the immunoglobulin genes for further characterization.

Major concerns

1. It is very difficult for readers to understand the logic how the whole scFv sequence could be extracted from the long read sequencing. The nature of scFv libraries used in this study was not described in this manuscript. And the extended Data Fig. 8 referred by authors as “describing the NGS protocol (Line 355)” is actually an irrelevant one. Two scFv libraries in reference 23 and 31 (Line 249) were referred to be used in this study but actually described in detail in reference 38 (PNAS 2015 112 p9967) of the reference 31 (J Biol Chem 2019 294 p19322). Based on the description of this reference, both scFv libraries are based on the humanized 4D5 (trastuzumab) framework. The first library varies CDRs L3, H1, H2, and H3 and describe in detail in the reference 38 (PNAS 2015 112 p9967) of reference 28 (J Biol Chem 2019 294 p19322). The second library was described to be varied in CDR L2 in addition to the other four CDRs and uses TRIM (trinucleotide mutagenesis) technology to generate a greater degree of diversity. But the nature of TRIM technology is not described in this manuscript or in reference 28. The extended Data Figure 1 is not providing information unsatisfactory to figure out how this long read sequencing is achieved. It can be speculated that the all the scFv genes in these scFv libraries share the same framework of humanized 4D5 and differ only in LCDR (2) and 3 and all HCDRs. If it is the case, long read sequencing can be achieved by primers based on framework regions. If this is the case, this strategy can be applied to only to synthetic scFv library constructed on the identical framework and cannot be used in other naïve, synthetic or immune scFv libraries with diverse panel of frameworks. This significantly limit the value of SLISY platform described in this manuscript. The authors are advised to clarify this point.

The reviewer is correct that our library is based on humanized 4D5 framework. Full length sequencing is applicable to any and all phage libraries provided that the backbone is known and contains constant regions. In other cases where the library contains minimal or no constant regions, the long reads from the end could be used to derive variable regions to be used as unique identifiers and PCR primers and an intermediate of PCR amplification could be used to derive the full length sequence. Therefore, while we make no claims in this

area, we expect this can include naïve, synthetic, or immune scFv libraries. We have also expanded and revised the text to further clarify how we sequence the full-length scFv as follows: "Given that our scFv library is based on the humanized 4D5 (trasuzumab) framework generated against ERBB2, we designed three sets of primers against various constant regions in the backbone so that three rounds of sequencing would cover all CDRs."

2. The authors used term "neutralize" in this manuscript. But the term "neutralizing activity of mAb" should be defined in in vitro or in vivo assay using SARS-CoV2 virus or at least pseudovirus. The authors actually employed competition ELISA.

We thank the reviewer for highlighting this important distinction and agree with clarification of language. We have now revised the text in multiple areas where we developed an in-house blocking ELISA for neutralization to instead say competition ELISA. In regards to results with the FDA-approved cPASS assay, because the test is designated for the detection of "neutralizing antibodies," we have continued to use the term neutralizing in this context (Taylor *et al.* J Clin Microbiol 2021) but have added qualifiers to make sure the reader understands that it is an *in vitro* assay (e.g., "using standardized *in vitro* assays").

Because the field of SARS-CoV-2 requires more extensive characterization of antibodies, we have now included infectivity assays with pseudovirus expressing the spike protein and luciferase. Briefly, we engineered HEK-293T cells to express recombinant human ACE2 protein. We then infected these cells with SARS-CoV-2 spike protein pseudoviruses containing luciferase. We then measured infectivity and blocking with antibodies by luciferase signal after 48 hours. To support our data that our antibodies will likely neutralize variants as well, we included pseudoviruses expressing the spike protein for the Beta, Delta, and Lambda variants. The results of these experiments are now included in the main text (Figs. 4e and 6c) and are summarized below:

The antibodies that we tested by competition ELISA and shown to be blocking did indeed show blocking against the pseudovirus. Of the three superclones identified by phage with the competition ELISA, two of them grafted well and continued to show neutralization of infectivity.

Reviewer #2 (Remarks to the Author):

Discovering recombinant antibodies is about to change thanks to ease of access to sequence data. But how to use that to actually get better antibodies? There is an apparent need to un-blackbox what is happening during selection and what is being enriched between multiple rounds of selection. The authors present a version of phage-ELISA coupled with NGS to link binding phenotype to sequence information that attempts to identify rare cross-reactive binders that are easily missed or might even get lost across multiple rounds of phage-display selection. This approach appears to tackle the often encountered “growth bias” quite nicely by sequencing the whole phage output and calculating a ratio of sequences encountered for the specific antigen against a control. It appears to be easily adaptable to other antigens and selection workflows but the authors should provide some more background on the phage library (such as diversity and the scaffold used). I have little doubt NGS will become much more widely adapted in phage-display panning campaigns and this study could serve as a template in how to use sequence information to prioritize candidate clones.

General comment: Occasionally the language seems sloppy, with interpretation of results appearing to be over-generalized from what is supported by the data. For example the anti-HLA phage display campaign could benefit from being treated more quantitatively (see specific comments below). It is unclear to me which main figure each of the extended data figures is related to (for example: Extended Data Figures 1, 2, 3, 4, 5, 6, 7 Extended Data Table 1, 2 all get referenced between Figure 1a and 1b) and the usage of figure captions instead as a supplementary results section (ex. Extended Data Figure 3) makes the manuscript occasionally very tedious to read.

L. 88-89 “universally amplify the CDR-H3 region of any scFv” - do those primers really work with any scFv from any library or what is the range of the “any” here?

We have clarified in the text to indicate "universally amplify the CDR-H3 region of our scFv library."

L. 95-96 “full length sequence of any phage” - not sure what that means. Do the authors mean “phagemid”? How long is the sequence of a phage?

We have clarified in the text to indicate "full length scFv sequence of each phagemid in our library."

L.98-99 “these methods (...) are described in detail in Materials and Methods” - seems to be rather unnecessary. I would assume that is where they would be described.

We have removed this sentence from the text as its meaning is implied and is redundant.

L. 104-105 “(...) desirable clones, which would be difficult to identify using a traditional growth based screening approach.” - please elaborate. It is not obvious what is considered desirable and “difficult to identify” is extremely hand-wavy. I have no doubt there are interesting things to be found, but this seems like the authors are guessing rather than measuring.

We agree and appreciate that the reviewer recognized one of main strengths of the SLISY, which is to identify and extract highly specific clones which are often at very low frequencies that would not be reasonable to randomly select by hand. We have revised and expanded the text in this section to better highlight this advantage. In the biopannings against both HLA-A3 and SARS-CoV-2, we observed that clones with the highest SBRs are at very low frequencies (Supplementary Table 1 and Figure 2c). We have summarized them below.

Starting Library	Validated Clones	SBR Round 4	Baseline Frequency	Round 4 Frequency	Estimated Probability of Isolating in 1,000 Random Clones
Nascent	HLAA3_1	44.00	3.81E-06	3.54E-05	0.03
Nascent	HLAA3_2	32.00	1.91E-06	2.48E-05	0.02
Nascent	HLAA3_3	29.25	9.53E-07	1.77E-04	0.16
Nascent	HLAA3_4	28.45	9.53E-07	3.89E-04	0.32
Nascent	HLAA3_5	21.00	9.53E-07	2.05E-04	0.19
Nascent	HLAA3_9	14.00	9.53E-07	1.06E-05	0.01
Nascent	HLAA3_6	13.50	9.53E-07	5.31E-05	0.05
Nascent	HLAA3_7	13.50	9.53E-07	9.91E-05	0.09
Nascent	HLAA3_8	10.21	9.53E-07	2.12E-03	0.88
A3-Clone20 spike-in	HLAA3_11	77.50	2.00E-06	8.16E-05	0.08
A3-Clone20 spike-in	HLAA3_5	36.55	2.00E-06	4.68E-04	0.37
A3-Clone20 spike-in	HLAA3_3	23.67	2.00E-06	2.93E-04	0.25
A3-Clone20 spike-in	HLAA3_4	23.04	2.00E-06	3.35E-04	0.28
A3-Clone20 spike-in	HLAA3_9	16.67	2.00E-06	1.24E-04	0.12
A3-Clone20 spike-in	HLAA3_10	14.40	2.00E-06	7.85E-05	0.08
A3-Clone20 spike-in	A3-Clone20	7.02	2.36E-05	6.78E-02	1.00

In traditional growth-based screening approaches, colonies are individually selected by random chance and tested for function. This process would be time-consuming and laborious. For example, the probability of isolating clone HLAA3_2 out of 1,000 random clones is 0.02. Furthermore, 7 of 11 HLA_A3 selected clones had a frequency less than 0.0001% in the final pool. If we picked and characterized a thousand clones, the approximate probability of isolating any given one of these 7 clones ranged from 0.01 to .09 and would be only 3E-10 for identifying all 7 in any one sampling.

Theoretically there is also no way for the investigator to know whether he/she has selected the most specific clones possible or not. Of course depending on downstream applications, one may ask if there is a lower limit for the specificity required for a desired antibody. However, SLISY provides a quantitative measure to increase one's chances to rapidly select desirable antibodies.

L. 109-110 “(...) improve both the number and quality of clones (...)” - again, considering this is the combined results/discussion section I am missing some sort of descriptive presentation of the obtained data. What is the “quality”, what is being improved there and by how much? “Improving the number” probably means more? How much more?

As described in comment directly above, the advantage of SLISY is the avoidance to select clones by random chance. With multiple rounds of biopanning, the goal is enrich for desired clones and increase chances of selecting those clones. However, we often see that good binders (those with highest SBRs) are still at very low frequencies that are not reliably realistic to obtain by random chance. We have expanded the text in that section to better highlight this point.

L.118-119 “Four rounds of traditional biopanning (...)” - Please be specific about what you mean with “traditional”.

"Traditional" was meant to indicate positive selection. However, we agree with the reviewer that its wording can be confusing and its implied meaning is redundant so we have removed this descriptor.

L. 125 “scFv clones with SBR > 10” - Why 10? Please explain why you choose this specific value as a cutoff.

We agree with the reviewer that the choice of the SBR cutoff was not made clear in the text. We have now included our reasoning in the text:

"Given that A3-Clone 20 as a positive control had an SBR of 7.02, we were interested in selecting clones with a higher SBR (>10) under the assumption that clones with at least a 10-fold theoretical specificity as detected by SLISY would be sufficient for validation."

L. 149 “a type of digital ELISA” - please explain further. What makes this assay digital?

We thank the reviewer for indicating further clarification of this term. Digital ELISAs are ELISA methods that detect single molecules which in this case are individual phagemids by sequencing. We have now elaborated in the text what we define as "digital" to avoid confusion as well as included a reference on digital ELISAs.

L. 178ff “Converting scFvs to full-length antibodies (...) can be problematic. We addressed this issue by converting the scFv regions (...) into full-length IgG antibodies.” - I would consider it

helpful if the authors could provide some detail into how conversion into IgG addresses the issues of conversion into IgG. What does “conversion” imply in this setting? Grafting CDR regions, cloning VH and VL?

We appreciate and agree with the reviewer about need to revise said statement. As the scFv presented on a phagemid is a significant different format than a full length soluble IgG antibody, it is not unexpected that there can be functional differences for the same scFv. We have now revised the sentences to better clarify that scFvs in phage format may not work in the more clinically useful IgG format.

L. 207 & L. 213 “ELISAs for millions of clones“ vs 1279 clones - I am confused. Did the authors test millions of clones using SLISY or did they test 1279? Could you not do ELISAs for the 1279 clones? Or were those derived from millions of clones that went into the assay?

We agree with the reviewer and wanted to clarify that the "millions of clones" mean the number of clones sequenced which is limited by the sequencing itself. This can be on the order of millions given that it is sequenced on a MiSeq and the library itself is much larger. To avoid confusion, we have eliminated any numerical descriptor.

L. 248 “Methods for scFv phage display library construction” - please provide a reference for the construction of the library used in this study. Ref 23 and Ref 31 refer themselves to another study (the same in this case). If the authors used the library from the Skora et al. paper, why not reference that? It is rather tiresome having to dig through another level of references within references.

We have corrected this now to indicate the original reference for the construction of the library.

Reviewer #3 (Remarks to the Author):

In their manuscript ("SLISY: Rapid identification of antibodies to evolving viral pathogens") the authors develop a novel strategy to identify binders by phage display by using differential enrichment after panning, in combination with high throughput sequencing. This is an interesting and innovative strategy, in principle suitable for publication in Nature Communications, and I greatly enjoyed reading the manuscript.

However, in its current form, I have two concerns with the paper:

1) The approach in many respects is reminiscent of the combinatorial scanning approach on phage developed at Genentech over 20 years ago (by Greg Weiss, Sachdev Sidhu and others). See: <https://www.pnas.org/doi/10.1073/pnas.160252097> and other publications. This needs to be extensively discussed and placed into historical context

We thank the reviewer for highlighting this important study utilizing shotgun scanning. We have now included it in the introduction of the main text. Although this study utilizes a phage-displayed library of alanine-substituted proteins to quickly identify functional

epitopes, as described it is only applicable to known binding partners. SLISY is a method for the *de novo* identification of highly specific antibodies that are selective against a target(s) versus a nontarget.

2) More importantly, the characterisation of the selected antibodies is limited to ELISA type assays. For the SARS-CoV-2 field in particular a more extensive characterisation using at least pseudoparticle assays, and preferably structural and vivo characterisation (human ACE-2 mice or hamster) would be expected. The assays outlined here would, in particular, not detect neutralising antibodies to important class 3 and 5 (non-ACE2) epitopes.

Please see response to Reviewer #1 Comment #2. We have now included infectivity assays with pseudovirus expressing the spike protein and luciferase. In addition to the spike protein of original Wuhan strain, we also included those for the Betta, Delta, and Lambda variants.

The reviewer brings up an interesting aspect about how neutralizing SARS-CoV-2 antibodies can be sorted into different classes based on where and how it binds to the spike protein (Deshpande *et al.* Front Immunol 2021). The purpose of this study highlights how SLISY can be used to rapidly identify highly selective antibodies against multiple targets including SARS-CoV-2. A significant number of scFvs that were converted to full-length antibodies continued to be neutralizing when tested against pseudoviruses. Therefore, by increasing the success rate for identifying selective antibodies, SLISY increases the chance it is neutralizing by default.

Although beyond the scope of this study, it is interesting to speculate about the nature of these neutralizing antibodies. Because the scFvs were selected based on purified recombinant spike protein, it is possible that the neutralizing antibodies bind to the RBD ACE2 binding site (Class 1), bind to the RBD in its up and down conformations while blocking ACE2 binding site (Class 2), and bind outside the ACE2 binding site on RBD (Class 3). Given that the phage was biopanned against purified spike protein, it would be unlikely to identify Class 4 neutralizing antibodies which bind to the spike protein once some large conformational change occurs. However one could theoretically identify these antibodies if biopanning was performed against a spike protein in this conformational change.

REVIEWERS' COMMENTS

Reviewer #2 (Remarks to the Author):

Overall, I am satisfied with the rebuttal. The findings are clearly presented and the additional experiments help make the conclusions stronger.

There is just one more minor comment regarding

L. 178ff "Converting scFvs to full-length antibodies (...) can be problematic. We addressed this issue by converting the scFv regions (...) into full-length IgG antibodies." -

The now revised sentence is: 'Converting scFvs presented on phagemids to full-length antibodies which are more suitable for clinical use can be problematic as there can be a lack of consistency between activity of the two formats. To evaluate whether the results described above are due to the scFv itself and independent of the phage structure, we converted the scFv regions from 12 of phage described above (eight neutralizing and four non-neutralizing) into full-length IgG 233 antibodies'

The revised sentence is still confusing. I think the part about 'conversion is problematic' could be dropped completely if what the authors want to address is 'using scFv as a therapeutic molecule can be problematic'.

Reviewer #3 (Remarks to the Author):

The author have addressed most of my concerns.

Figure 4e:

- Please provide proper IC50 values and serial dilutions (potentially through CRO or collaboration).

REVIEWERS' COMMENTS

Reviewer #2 (Remarks to the Author):

Overall, I am satisfied with the rebuttal. The findings are clearly presented and the additional experiments help make the conclusions stronger.

There is just one more minor comment regarding

L. 178ff "Converting scFvs to full-length antibodies (...) can be problematic. We addressed this issue by converting the scFv regions (...) into full-length IgG antibodies." -

The now revised sentence is: 'Converting scFvs presented on phagemids to full-length antibodies which are more suitable for clinical use can be problematic as there can be a lack of consistency between activity of the two formats. To evaluate whether the results described above are due to the scFv itself and independent of the phage structure, we converted the scFv regions from 12 of phage described above (eight neutralizing and four non-neutralizing) into full-length IgG 233 antibodies'

The revised sentence is still confusing. I think the part about 'conversion is problematic' could be dropped completely if what the authors want to address is 'using scFv as a therapeutic molecule can be problematic'.

We thank the reviewer for the opportunity to allow us to clarify this potentially confusing sentence. We agree that the way the sentence was written in the original text did not properly convey the message that we wanted. The conversion of scFvs presented on phagemids to full length antibodies is problematic as phage, which are very large relative to its scFv, can potentially cause steric issues that may ultimately exhibit downstream blocking effects which are not directly related to the scFv itself. Therefore, functional assays with scFv-expressing-phage may in theory produce misleading effects.

Direct comparisons regarding the therapeutic use of scFvs or full length antibodies are beyond the scope of this study. As full length antibodies are the most commonly used format, we have now revised the text by removing that initial sentence and adding "which are more suitable for clinical use" at the end of the second sentence:

"To evaluate whether the results described above are due to the scFv itself and independent of the phage structure, we converted the scFv regions from 12 of the phage described above (eight neutralizing and four non-neutralizing) into full-length IgG antibodies, which are more suitable for clinical use (Fig. 4a).³³"

Reviewer #3 (Remarks to the Author):

The author have addressed most of my concerns.

Figure 4e:

- Please provide proper IC50 values and serial dilutions (potentially through CRO or collaboration).

We thank the reviewer for highlighting the importance of characterizing the relationship between isolated antibodies and their antigens. The IC50 represents the inhibition at which 50% of the antibody is bound and 50% is not bound. This value can range widely depending on the specific set up of the inhibition assay and therefore is not considered a constant value (Barbet *et al.* Pharm Stat 2019). While these values can be helpful in determining suitability for downstream therapeutic design, this is beyond the scope of this study as we are not making any direct claims regarding therapeutic potential.

However, we did include K_d values for our converted full-length antibodies. A calculated ratio of K_{off}/K_{on} , the K_d in this study is the equilibrium dissociation constant between the antibody and its antigen (SARS-CoV-2 spike protein). K_d values are generally considered a more intrinsic value as they are more independent from the assay. They vary less with changes in concentration of antigen and antibody. The Surface Plasmon Resonance (SPR) data highlights that our antibodies identified by SLISY have high affinity to the SARS-CoV-2 spike protein and is summarized in Supplementary Table 4 and below:

Antibody (Ligand)	Protein (Analyte)	k_a (1/Ms)	k_d (1/s)	K_D (nM)	Chi ²	U-Value	RU
RBD_1	SARS-CoV-2 FL	0.840×10^4	3.710×10^{-4}	44.2	1.35	3	~30
S1_13	SARS-CoV-2 FL	1.578×10^4	1.840×10^{-4}	11.7	0.168	1	~60
FL_5	SARS-CoV-2 FL	1.146×10^4	4.550×10^{-4}	39.7	0.518	2	~20
FL_10	SARS-CoV-2 FL	1.396×10^4	1.048×10^{-4}	7.5	1.34	3	~120
FL_12	SARS-CoV-2 FL	1.489×10^4	8.860×10^{-5}	6	3.5	5	~120
FL_13	SARS-CoV-2 FL	1.077×10^4	1.022×10^{-4}	9.5	1.23	4	~100

Editor:

Code and Software Submission Checklist

During peer review we did not realise that your work involved the use of custom code. We should have requested that the code be made available to reviewers, and should have requested a completed Code and Software Submission Checklist. We would now like to clarify if and how the software/algorithms necessary to reproduce the results will be made available to the scientific community upon publication as required by our material sharing requirements.

We thank the editor for catching this omission and have addressed it as follows:

- 1) We have now completed and supplied a Code and Software Submission Checklist.**
- 2) We have made the code available through Zenodo with the following URL: <https://doi.org/10.5281/zenodo.7154344>**
- 3) We have edited the text to more accurately describe the custom part of the software as follows.**

Old Text: “Sequences processed and translated using a custom SQL database and both the nucleotide sequences and amino acid translations were analyzed using Microsoft Excel.”

New Text “Sequences were demultiplexed and processed to extract and translate CDRH3 regions using custom python software (<https://doi.org/10.5281/zenodo.7154344>) and analyzed using SQL databases (MSSQL) and Microsoft Excel.”

- 4) Finally, we would note that the code was not integral to the evaluation or practice of SLISY and could easily be implemented or recreated by individuals with experience working with NGS data.**